# WILTing Trees: Interpreting the Distance Between MPNN Embeddings

### Abstract

We investigate the distance function implicitly learned by message passing neural networks (MPNNs) on specific tasks. Our goal is to capture the *functional* distance that is implicitly learned by an MPNN for a given task. This contrasts previous work which relates MPNN distances on arbitrary tasks to *structural* distances that ignore the task at hand. To this end, we distill the distance between MPNN embeddings into an interpretable graph distance. Our distance is an optimal transport on the Weisfeiler Leman Labeling Tree (WILT), whose edge weights reveal subgraphs that strongly influence the distance between MPNN embeddings. Moreover, it generalizes the metrics of two well-known graph kernels and is computable in linear time. Through extensive experiments, we show that MPNNs define the relative position of embeddings by focusing on a small number of subgraphs known by domain experts to be functionally important.

## 1 Introduction

Message passing graph neural networks (MPNNs) have been reported to achieve high predictive performance in various domains (Zhou et al., 2020). To understand these performance gains, many studies have focused on the expressive power of MPNNs (Morris et al., 2019; Xu et al., 2019; Maron et al., 2019). However, the binary nature of expressive power excludes any analysis of the distance between graph embeddings, which is considered to be a key to the predictive power of MPNNs (Liu et al., 2022b; Li & Leskovec, 2022; Morris et al., 2024). Recently, there has been growing interest in the analysis of MPNN (generalization) performance using *structural* distances between graphs Böker et al. (2024); Franks et al. (2024) that consider graph topology but ignore the target function to be learned. One can then derive generalization bounds under assumptions on the margin between classes or on Lipschitz constants of the target function. Both assumptions do often not hold on real data and MPNN architectures used in practice. In this work, we instead investigate the distance $d_{\text{MPNN}}$ implicitly obtained from an MPNN and its relation to a *functional* distance $d_{\text{func}}$ defined on the target values of the learning task.

Specifically, we ask: *What properties does the distance $d_{MPNN}$ learned by a well-performing MPNN have in practice that can explain the high performance?* While previous studies (Chuang & Jegelka, 2022; Böker et al., 2024) focused on the alignment between $d_{\text{MPNN}}$ and a non-task-tailored structural graph distance $d_{\text{struc}}$, we have found that it is not critical to the predictive performance of MPNNs. Rather, even if an MPNN was trained with classical cross-entropy loss, $d_{\text{MPNN}}$ respects the task-relevant functional distance $d_{\text{func}}$ and the alignment between both is highly correlated with the predictive performance of MPNNs. Then, we move to our second question: *How do MPNNs learn such a metric structure?* Since MPNNs essentially consider graphs as multisets of Weisfeiler Leman (WL) subgraphs, we propose a method to identify WL subgraphs whose presence in a graph significantly affects its relative position to other graphs in the MPNN embedding space. Specifically, we distill MPNNs into a *weighted Weisfeiler Leman Labeling Tree* (WILT) while preserving $d_{\text{MPNN}}$. The WILT yields an optimal transport distance on a tree ground metric, which we prove to be a trainable generalization of the graph distances of existing high-performance kernels (Kriege et al., 2016; Togninalli et al., 2019). We show experimentally that the WILTing tree distance fits MPNN distances well. Examination of the resulting edge parameters on WILT after distillation shows that only a small number of WL subgraphs determine $d_{\text{MPNN}}$. In a qualitative experiment, the subgraphs that strongly influence $d_{\text{MPNN}}$ are those that are known to be functionally important by domain knowledge. In short, our contributions are:

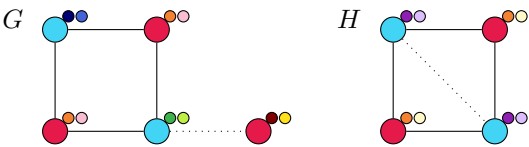

Figure 1: Examples of how the Weisfeiler Leman algorithm works on graphs. ● and ● are colors corresponding to initial node labels. Node colors in iterations one and two are shown in the small circles next to the nodes. For example, ○ $= (\bullet, \{\!\{(\circ, -), (\circ, -)\}\!\})$ and ○ $= (\bullet, \{\!\{(\circ, -), (\circ, -), (\bullet, \cdots)\}\!\})$.

- We show that MPNN distances after training are aligned with the task-relevant functional distance of the graphs and that this is key to the high predictive performance of MPNNs.
- We propose a trainable graph distance on a weighted Weisfeiler-Lehman Labeling Tree (WILT) that generalizes Weisfeiler Leman-based distances and is efficiently computable.
- WILTs allow a straightforward definition of *relevant* subgraphs. Thus, distilling an MPNN into a WILT enables us to identify subgraphs that strongly influence the distance between MPNN embeddings, allowing an interpretation of the MPNN embedding space.

## 2   PRELIMINARIES

We define a graph as a tuple $G = (V, E, l_{\text{node}}, l_{\text{edge}})$, where $V$ and $E$ are the set of nodes and edges, respectively. Each node and each edge have a label defined by $l_{\text{node}} : V \to \Sigma_{\text{node}}$ and $l_{\text{edge}} : E \to \Sigma_{\text{edge}}$, where $\Sigma_{\text{node}}$ and $\Sigma_{\text{edge}}$ are finite sets. We restrict them to finite sets because our method is based on the Weisfeiler Leman test described below, which is discrete in nature. We denote the set of all labeled graphs up to isomorphism as $\mathcal{G}$. Note that we only consider undirected graphs, but extending our work to directed graphs is easy by employing an appropriate version of the Weisfeiler Leman test. We denote the set of neighbors of node $v$ as $\mathcal{N}(v)$.

**Message Passing Algorithms** (Gilmer et al., 2017) include popular GNNs such as Graph Convolutional Networks (GCN, Kipf & Welling, 2017), and Graph Isomorphism Networks (GIN, Xu et al., 2019). At each iteration, a message passing algorithm updates the embeddings of all nodes by aggregating the embeddings of themselves and their neighbors in the previous iteration. After $L$ iterations, the node embeddings are aggregated into the graph embedding $h_G$:

$$h_v^{(l)} = \text{UPD}^{(l)}\left(h_v^{(l-1)}, \text{AGG}^{(l)}\left(\{\!\{(h_u^{(l-1)}, e_{vu}) \mid u \in \mathcal{N}(v)\}\!\}\right)\right) \quad h_G = \text{READ}\left(\{\!\{h_v^{(L)} \mid v \in V\}\!\}\right)$$

Here $\{\!\{\}\!\}$ denotes a multiset and $0 < l \le L$ with $h_v^{(0)} = l_{\text{node}}(v)$. $h_v^{(l)} \in \mathbb{R}^d$ and $h_G \in \mathbb{R}^{d'}$ are the embedding of node $v$ after the $l$-th layer and the graph embedding, respectively. $\text{AGG}^{(l)}$, $\text{UPD}^{(l)}$, and READ are functions. *Message Passing Neural Networks* (MPNNs) implement $\text{UPD}^{(l)}$ and $\text{AGG}^{(l)}$ using multilayer perceptrons (MLPs). Sum and mean pooling are popular for READ.

**The Weisfeiler Leman (WL) Algorithm** is a message passing algorithm, where $\text{UPD}^{(l)}$ is an injective function. $\text{AGG}^{(l)}$ and READ are the identity function on multisets. A node embedding of the WL algorithm is called *color*. We use $c_v^{(l)}$ instead of $h_v^{(l)}$ to refer to it. Figure 1 shows the progress of the WL algorithm on two graphs: $G$ and $H$ start with the same colors, but after two iterations, they no longer share any colors, i.e., $\{\!\{c_v^{(2)} \mid v \in V_G\}\!\} \cap \{\!\{c_v^{(2)} \mid v \in V_H\}\!\} = \emptyset$.

**Message Passing Pseudometrics** The WL algorithm cannot distinguish some nonisomorphic graphs (Cai et al., 1992) and all MPNNs are bounded by its expressiveness (Xu et al., 2019). Hence, any MPNN yields a pseudometric on the set of pairwise nonisomorphic graphs $\mathcal{G}$.

**Definition 1** (Graph Pseudometric)**.** *A graph pseudometric space $(\mathcal{G}, d)$ is given by a non-negative real valued function $d : \mathcal{G} \times \mathcal{G} \to \mathbb{R}_{\ge 0}$ that satisfies for all $F, G, H \in \mathcal{G}$:*

$$d(G, G) = 0 \qquad \textit{(Identity)}$$
$$d(G, H) = d(H, G) \qquad \textit{(Symmetry)}$$
$$d(G, F) \le d(G, H) + d(H, F) \qquad \textit{(Triangle inequality)}$$

Given an MPNN, we obtain a pseudometric space $(\mathcal{G}, d_{\text{MPNN}})$ by setting $d_{\text{MPNN}}(G, H) := d(h_G, h_H)$, where $d : \mathbb{R}^{d'} \times \mathbb{R}^{d'} \to \mathbb{R}$ is a (pseudo)metric and $h_G$ and $h_H$ are graph embeddings. Note that $(\mathcal{G}, d_{\text{MPNN}})$ is not a metric space since there are nonisomorphic graphs $G, H$ with identical representations and hence $d_{\text{MPNN}}(G, H) = 0$. For the remainder of this paper, we will use $d_{\text{MPNN}}(G, H) = ||h_G - h_H||_2$, but other distances between embeddings can also be used. It should be noted that $d_{\text{MPNN}}$ depends not only on the input graphs but also on the task on which the MPNN is trained. For example, $d_{\text{MPNN}}$ of an MPNN trained to predict the toxicity of molecules will be different from the $d_{\text{MPNN}}$ of another MPNN trained to predict the solubility of the same molecules.

**Structural Pseudometrics** To date, many different graph kernels have been proposed (Kriege et al., 2020). Each positive semidefinite graph kernel $k : \mathcal{G} \times \mathcal{G} \to \mathbb{R}$ corresponds to a pseudometric between graphs. See Appendix B.1 for how the kernels used in this article are transformed into corresponding pseudometrics. We will refer to these pseudometrics as *structural* pseudometrics and write $d_{\text{struc}}$, as they only consider the structural and node/edge label information of graphs, without being trained using the target label information on a training set.

**Functional Pseudometrics** To formally define the functional distance between graphs, we introduce another pseudometric on $\mathcal{G}$ that is based on the target labels of the graphs.

**Definition 2** (Functional Pseudometric). *Let $y_G$ be the target label of graph $G$ in a given task. In classification, $y_G$ is a categorical class, while $y_G$ is a numerical value in regression. We assume the space for $y_G$ is bounded. Then, the functional pseudometric space $(\mathcal{G}, d_{\text{func}})$ is obtained from $d_{\text{func}} : \mathcal{G} \times \mathcal{G} \to [0, 1]$ defined as:*

$$d_{\text{func}}(G, H) := \begin{cases} \mathbb{1}_{y_G \neq y_H} & (\textit{classification}) \\ \frac{|y_G - y_H|}{\sup\limits_{I \in \mathcal{G}} y_I - \inf\limits_{I \in \mathcal{G}} y_I} & (\textit{regression}), \end{cases}$$

*where $\mathbb{1}_{y_G \neq y_H}$ is the indicator function that returns 1 if $y_G \neq y_H$, otherwise 0.*

See Appendix B.2 for a proof that $(\mathcal{G}, d_{\text{func}})$ is a pseudometric space. If the sup/inf of $y_G$ in $\mathcal{G}$ are unknown, they can be approximated by the max/min in a training dataset $\mathcal{D}$.

**The Expressive Power** of a message passing algorithm is defined based on its ability to distinguish non-isomorphic graphs. Formally, a message passing graph embedding function $f$ is said to be at least as expressive as another one $g$ if the following holds:

$$\forall G, H \in \mathcal{G} : f(G) = f(H) \implies g(G) = g(H),$$

where $\mathcal{G}$ is the set of all pairwise non-isomorphic graphs. We extend the above to pseudometrics on graphs. Specifically, a graph pseudometric $d$ is said to be at least as expressive as $d'$ $(d \geq d')$ iff

$$\forall G, H \in \mathcal{G} : d(G, H) = 0 \implies d'(G, H) = 0.$$

$d$ and $d'$ are equally expressive $(d \cong d')$ iff $d \geq d'$ and $d' \geq d$. Furthermore, $d$ is said to be more expressive than $d'$ $(d > d')$ iff $d \geq d'$ and there exists $G, H \in \mathcal{G}$ s.t. $d(G, H) \neq 0 \land d'(G, H) = 0$.

## 3 Is the MPNN Embedding Distance Critical to Performance?

Our first question is what properties $d_{\text{MPNN}}$ of well performing MPNNs have in practice that can explain their high performance. This section investigates whether the alignment between $d_{\text{MPNN}}$ and the *task-relevant* pseudometric $d_{\text{func}}$ is such a property. Specifically, we address questions below:

**Q1.1** Does training MPNN increase the alignment between $d_{\text{MPNN}}$ and the task-relevant $d_{\text{func}}$?

**Q1.2** Does a strong alignment between $d_{\text{MPNN}}$ and $d_{\text{func}}$ indicate high performance of the MPNN?

Note that the alignment between $d_{\text{MPNN}}$ and *task-irrelevant* structural graph pseudometrics $d_{\text{struc}}$ has been considered a key to MPNN performance in previous studies (Chuang & Jegelka, 2022; Böker et al., 2024; Franks et al., 2024). However, we found that this property is not consistently improved by training and does not correlate with performance. (See Appendix E for detailed analyses).

To answer **Q1.1** and **Q1.2**, we should first define a measure of the alignment between $d_{\text{MPNN}}$ and $d_{\text{func}}$. Note that it is inappropriate to adopt a typical min/max of $\frac{d_{\text{func}}(G, H)}{d_{\text{MPNN}}(G, H)}$ to measure the alignment. This is because $d_{\text{func}}$ is a binary function for classification tasks, and expecting the exact match of the two distances is unreasonable. Thus, we define another evaluation criterion in the following.

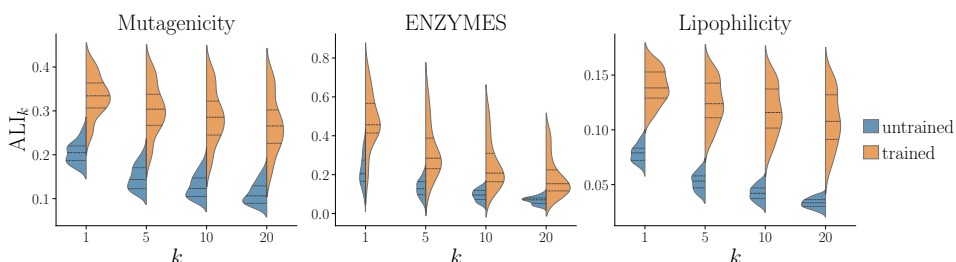

Figure 2: The distribution of $\text{ALI}_k(d_{\text{MPNN}}, d_{\text{func}})$ under different $k$ and datasets.

Table 1: Correlation between $\text{ALI}_k(d_{\text{MPNN}}, d_{\text{func}})$ and the performance on $\mathcal{D}_{\text{train}}$ and $\mathcal{D}_{\text{test}}$ under different $k$. Performance is accuracy for Mutagenicity and ENZYMES, and RMSE for Lipophilicity.

| | Mutagenicity | | | | ENZYMES | | | | Lipophilicity | | | |
|---|---|---|---|---|---|---|---|---|---|---|---|---|
| k | 1 | 5 | 10 | 20 | 1 | 5 | 10 | 20 | 1 | 5 | 10 | 20 |
| train | 0.71 | 0.69 | 0.67 | 0.64 | 0.88 | 0.81 | 0.77 | 0.74 | -0.74 | -0.72 | -0.70 | -0.69 |
| test | 0.71 | 0.69 | 0.66 | 0.64 | 0.49 | 0.43 | 0.38 | 0.34 | -0.56 | -0.53 | -0.53 | -0.52 |

**Definition 3** (Evaluation Criterion for Alignment Between $d_{\text{MPNN}}$ and $d_{\text{func}}$). *Let $\mathcal{D}$ be a graph dataset, $k$ be an integer hyperparameter, and $\mathcal{N}_k(G) \subset \mathcal{D} \setminus \{G\}$ be a set of $k \geq 1$ graphs that are closest to $G$ under $d_{\text{MPNN}}$. Let*

$$A_k(G) \coloneqq \frac{1}{k} \sum_{H \in \mathcal{N}_k(G)} d_{\text{func}}(G, H), \quad B_k(G) \coloneqq \frac{1}{|\mathcal{D}| - k - 1} \sum_{H \in \mathcal{D} \setminus (\mathcal{N}_k(G) \cup \{G\})} d_{\text{func}}(G, H).$$

*Then $d_{MPNN}$ is aligned with $d_{func}$ if*

$$\text{ALI}_k(d_{MPNN}, d_{func}) \coloneqq \frac{1}{|\mathcal{D}|} \sum_{G \in \mathcal{D}} \left[ -A_k(G) + B_k(G) \right]$$

*is positive. In addition, the larger $ALI_k$ is, the more we say $d_{MPNN}$ is aligned with $d_{func}$.*

Here, $A_k(G)$ and $B_k(G)$ are the average functional distances between $G$ and its neighbors and non-neighbors, respectively. If $A_k(G) < B_k(G)$, then the MPNN embeds $G$ and functionally similar graphs closer on average than functionally dissimilar graphs.

We show the distribution of $\text{ALI}_k(d_{\text{MPNN}}, d_{\text{func}})$ for 48 different MPNNs on different datasets and varying $k$ in Figure 2. Each model was trained with a standard loss function (cross entropy loss for classification and RMSE for regression). We did not explicitly optimize $\text{ALI}_k$. We also include the results for untrained MPNNs to see the effect of training. We can see that there is little overlap between the distributions of the untrained and trained MPNNs. This means that $\text{ALI}_k$ consistently improves a lot through training, implying a positive answer to **Q1.1**. Next, we tested the Pearson correlation coefficient (PCC) between $\text{ALI}_k(d_{\text{MPNN}}, d_{\text{func}})$ of trained MPNNs and their predictive performance. We use accuracy and RMSE between the ground truth target and predicted values to measure classification and regression performance, respectively. Table 1 shows that the PCC for Mutagenicity and ENZYMES is close to one, indicating that the higher the $\text{ALI}_k$, the higher the accuracy. Similarly, the higher the $\text{ALI}_k$, the lower the RMSE for Lipophilicity. The correlations are consistent across training and test sets. These results suggest that the degree of alignment between $d_{\text{MPNN}}$ and $d_{\text{func}}$ is a crucial factor contributing to the high performance of MPNNs, answering **Q1.2** positively. See Appendix D for more details and additional results on non-molecular datasets.

## 4 WILTING PSEUDOMETRICS

Section 3 confirms that MPNNs are implicitly trained so that $d_{\text{MPNN}}$ aligns with $d_{\text{func}}$, which turns out to be crucial for MPNN's performance. Then, our second research question is: How do MPNNs

learn $d_{\text{MPNN}}$ that respects $d_{\text{func}}$? Since MPNN embeddings are aggregations of WL color embeddings, we can infer that MPNNs learn during training which WL colors are important for capturing the task-relevant functional graph distance $d_{\text{func}}$. This determines the relative position between MPNN embeddings based on the existence of such WL colors in graphs. To identify WL colors that strongly influence $d_{\text{MPNN}}$, we propose to distill $d_{\text{MPNN}}$ into our new graph pseudometric $d_{\text{WILT}}$, which has tunable weights and is based on the WL colors of the input graphs. $d_{\text{WILT}}$ is an optimal transport distance on the Weisfeiler Leman Labeling Tree (WILT) and generalizes two existing distances of high-performing graph kernels (Kriege et al., 2016; Togninalli et al., 2019). After distillation, the parameters of $d_{\text{WILT}}$ allow us to identify WL colors that are considered important by the MPNN.

## 4.1 WEISFEILER LEMAN LABELING TREE (WILT)

The Weisfeiler Leman Labeling Tree (WILT) $T_{\mathcal{D}}$ is a rooted weighted tree built from the set of colors obtained by the WL test on a graph dataset $\mathcal{D} \subseteq \mathcal{G}$. Given $\mathcal{D}$, we define $V(T_{\mathcal{D}})$ as the *set* of colors that appear on any node during the WL test plus the root node $r$, that is, $V(T_{\mathcal{D}}) = \{c_v^{(l)} \mid v \in V_G, G \in \mathcal{D}, l \in [L]\} \cup \{r\}$. Colors $x, y \in V(T_{\mathcal{D}}) \setminus \{r\}$ are connected in $T_{\mathcal{D}}$ if and only if there exists a node $v$ in some graph in $\mathcal{D}$ and an iteration $l$ with $x = c_v^{(l)}$ and $y = c_v^{(l-1)}$. $r$ is connected to all $x = c_v^{(0)}$. Due to the injectivity of the AGG and UPD functions in the WL algorithm, it follows that $T_{\mathcal{D}}$ is a tree. Figure 3 (upper left) shows the WILT built from the graphs $G$ and $H$ in Figure 1. See Appendix C for a detailed algorithm to build a WILT from $\mathcal{D}$.

We consider edge weights $w : E(T_{\mathcal{D}}) \to \mathbb{R}_{\geq 0}$ on WILT. We only allow non-negative weights so that the WILTing distance in Definition 4 will be non-negative. Given a WILT $T_{\mathcal{D}}$ with weights $w$, the shortest path distance $d_{\text{path}}(x, y; w) := \sum_{e \in \text{Path}(x,y)} w(e)$ is the sum of edge weights of the unique shortest path $\text{Path}(x, y)$ between $x$ and $y$. Note that $d_{\text{path}}$ is a pseudometric on $V(T_{\mathcal{D}})$, i.e., the set of WL colors in $\mathcal{D}$. Intuitively, $d_{\text{path}}(x, y; w)$ is large if $\text{Path}(x, y)$ is long, but $w$ allows us to tune this distance according to the needs of the learning task.

## 4.2 THE WILTING DISTANCE

A WILT $T_{\mathcal{D}}$ with edge weights $w$ yields a pseudometric $d_{\text{WILT}}$ on the graph set $\mathcal{D}$. This section shows two equivalent characterizations of $d_{\text{WILT}}$ as an optimal transport distance and as a weighted Manhattan distance. The latter allows us to define the importance of specific WL colors and to compute our proposed distance efficiently. For simplicity, we define $d_{\text{WILT}}$ for graphs with the same number of nodes. In the next section, we will discuss the extension to graphs with different numbers of nodes. For two distributions with identical mass on the same pseudometric space, optimal transport distances such as the Wasserstein distance (Villani, 2009) measure the minimum effort of shifting probability mass from one distribution to the other. Each unit of shifted mass is weighted by the distance it is shifted. We define our pseudometric $d_{\text{WILT}}(G, H; w)$ as the optimal transport between $V_G$ and $V_H$, where the ground pseudometric is the shortest path metric on the WILT $T_{\mathcal{D}}$.

**Definition 4** (WILTing Distance). *Let $G, H \in \mathcal{D}$ be graphs with $|V_G| = |V_H|$. Then*

$$d_{\text{WILT}}(G, H; w) := \min_{P \in \Gamma} \sum_{v_i \in V_G} \sum_{u_j \in V_H} P_{i,j} d_{\text{path}}(c_{v_i}^{(L)}, c_{u_j}^{(L)})$$

*where $\Gamma := \{P \in \mathbb{R}^{|V_G| \times |V_H|} \mid P_{i,j} \geq 0, P\mathbf{1} = \mathbf{1}, P^T\mathbf{1} = \mathbf{1}\}$.*

Note that $d_{\text{WILT}}$ is not a metric but a pseudometric on the set of pairwise nonisomorphic graphs $\mathcal{G}$. This is because there are nonisomorphic graphs $G$ and $H$ whose colors are the same after $L$ iterations, i.e., $\{\!\{c_v^{(L)} \mid v \in V_G\}\!\} = \{\!\{c_v^{(L)} \mid v \in V_H\}\!\}$.

Generic algorithms to compute Wasserstein distances require cubic runtime. In our case, however, there exists a *linear* time algorithm to compute $d_{\text{WILT}}$ as shown below, since the ground pseudometric $d_{\text{path}}$ is the shortest path metric on a tree (Le et al., 2019).

**Definition 5** (WILT Embedding). *The WILT embedding of a graph $G \in \mathcal{D}$ is a vector, where each dimension counts how many times a corresponding WL color appears during the WL test on $G$, i.e., $\nu_c^G := |\{v \in V_G \mid \exists l \in [L]\, c_v^{(l)} = c\}|$ for $c \in V(T_{\mathcal{D}}) \setminus \{r\}$. (see upper right of Figure 3).*

Figure 3: (upper left): The Weisfeiler Leman Labelling Tree (WILT) built from $\mathcal{D} = \{G, H\}$ from Figure 1. (lower left): The WILT built from $\mathcal{D} = \{G, H\}$ with dummy nodes. (right): The WILT embeddings $\nu$, $\dot{\nu}$ with size normalization, and $\bar{\nu}$ with dummy node normalization.

**Proposition 1** (Equivalent Definition of WILTing Distance). *$d_{WILT}$ in Definition 4 is equivalent to:*

$$d_{WILT}(G, H; w) = \sum_{c \in V(T_\mathcal{D}) \backslash \{r\}} w\left(e_{\{c, p(c)\}}\right) \left| \nu_c^G - \nu_c^H \right|,$$

*where $e_{\{c, p(c)\}}$ is the edge connecting $c$ and its parent $p(c)$ in $T_\mathcal{D}$.*

This equivalence allows efficient computation of $d_{\text{WILT}}$ given the WILT embeddings of graphs, which can be computed by the WL algorithm in $O(L|E_G|)$ time, where $L$ is the number of WL iterations. Using sparse vectors for $\nu^G$ and $\nu^H$, $d_{\text{WILT}}(G, H)$ can be computed in $O(|V_G| + |V_H|)$.

### 4.3 NORMALIZATION AND SPECIAL CASES OF WILTING DISTANCE

The definition of $d_{\text{WILT}}(G, H)$ as an optimal transport distance requires $|V_G| = |V_H|$. However, $|V_G|$ and $|V_H|$ usually do not match, so we propose two solutions. Interestingly, the two modified WILTing distances generalize two distance functions corresponding to well-known graph kernels.

**Size Normalization** Straightforwardly, we can restrict the mass of each node to $\frac{1}{|V_G|}$ when calculating the Wasserstein distance in Definition 4. In other words, we replace $\Gamma$ with $\dot{\Gamma} := \{P \in \mathbb{R}^{|V_G| \times |V_H|} \mid P_{i,j} \geq 0, P\mathbf{1} = \frac{1}{|V_G|}\mathbf{1}, P^T\mathbf{1} = \frac{1}{|V_H|}\mathbf{1}\}$. Similarly, $\nu^G$ in Proposition 1 is changed to $\dot{\nu}^G := \frac{\nu^G}{|V_G|}$. The resulting distance $\dot{d}_{\text{WILT}}$ effectively ignores differences in the number of nodes of $G$ and $H$, generally assigning fractions of colors in $G$ to colors in $H$. In Figure 3 (right center), we show $\dot{\nu}$ of $G$ and $H$ in Figure 1. $\dot{d}_{\text{WILT}}(G, H)$ is calculated as:

$$\dot{d}_{\text{WILT}}(G, H) = w(e_{\{\bullet, \bigcirc\}}) \left| \frac{3}{5} - \frac{2}{4} \right| + w(e_{\{\bullet, \bigcirc\}}) \left| \frac{2}{5} - \frac{2}{4} \right| + \ldots + w(e_{\{\bullet, \bigcirc\}}) \left| \frac{0}{5} - \frac{2}{4} \right|.$$

An interesting property of $\dot{d}_{\text{WILT}}$ is that it generalizes the pseudometric corresponding to the Wasserstein Weisfeiler Leman graph kernel (Togninalli et al., 2019): When $w \equiv \frac{1}{2(L+1)}$, $\dot{d}_{\text{WILT}}$ matches their distance. See Appendix B.3 for technical details.

**Dummy Node Normalization** We can also add isolated nodes with a special label, called dummy nodes, to graphs so that all the graphs have the same number of nodes. The WILT will be built in the same way as described in Section 4.1 after dummy nodes are added to all graphs in $\mathcal{D}$. The resulting WILT has new colors $c_\neg^0, c_\neg^1, \ldots, c_\neg^L$ that arise from the WL iteration on the isolated dummy nodes (Figure 3 lower left). The WILT embedding will be slightly changed to

$$\bar{\nu}_c^G := \begin{cases} N - |V_G| \text{ if } c \in \{c_\neg^0, c_\neg^1, \ldots, c_\neg^L\} \\ \nu_c^G \text{ otherwise} \end{cases},$$

---

**Algorithm 1** Optimizing edge weights of WILT

---

**Input**: Graph dataset $\mathcal{D}$, an MPNN $f$ with $L$ message passing layers trained on $\mathcal{D}$, and WILT $T_{\mathcal{D}}$ built from the results of $L$-iteration WL test on $\mathcal{D}$
**Parameter**: batch size, number of epochs $E$, and learning rate $lr$
**Output**: Optimized edge weights of WILT $T_{\mathcal{D}}$

    $n_c \leftarrow |E(T_{\mathcal{D}})|$
    $w \leftarrow \mathbb{1} \in \mathbb{R}^{n_c}$
    optimizer $\leftarrow$ Adam(params=$w$, lr=$lr$)
    **for** $e = 1$ to $E$ **do**
        **for** batch $B$ in $\mathcal{D}^2$ **do**
            $l \leftarrow \frac{1}{|B|} \sum\limits_{(G,H) \in B} \Big( d_{\text{WILT}}(G, H) - d_{\text{MPNN}}(G, H) \Big)^2$
            $l$.backward()
            optimizer.step()
            $w \leftarrow \max(w, 0)$             $\triangleright$ Ensuring that each dimension of $w$ is non-negative
        **end for**
    **end for**
    **return** $w$

---

where $N = \max_{G \in \mathcal{D}} |V_G|$ (See Figure 3 (lower right)). Then, the resulting distance $\bar{d}_{\text{WILT}}(G, H)$ for the graphs in Figure 1 is:

$$\bar{d}_{\text{WILT}}(G, H) = w(e_{\{\bullet, \circ\}})|3 - 2| + w(e_{\{\bullet, \circ\}})|2 - 2| + \ldots + w(e_{\{\circ, \circ\}})|0 - 1|.$$

Similar to size normalization, $\bar{d}_{\text{WILT}}$ includes the pseudometric of Weisfeiler Leman optimal assignment kernel (Kriege et al., 2016) as a special case. When $w \equiv \frac{1}{2}$, $\bar{d}_{\text{WILT}}$ is equivalent to their distance. See Appendix B.3 for more details.

## 4.4 WILTing Tree Learning and Identification of Important WL Colors

Now, we have a graph distance on WILT defined for any pairs of graphs in $\mathcal{D}$. Next, we show how to optimze the edge weights $w$. Proposition 1 allows us to learn the edge weights $w$, given training data. Specifically, given a target distance $d_{\text{target}}$ we adapt the distance function $d_{\text{WILT}}$ by minimizing

$$\mathcal{L}(w) \coloneqq \sum_{(G,H) \in \mathcal{D}^2} \Big( d_{\text{WILT}}(G, H; w) - d_{\text{target}}(G, H) \Big)^2,$$

with respect to $w$. Note that $d_{\text{WILT}}$ can refer to both $\dot{d}_{\text{WILT}}$ and $\bar{d}_{\text{WILT}}$. In this work, we focus on $d_{\text{target}} = d_{\text{MPNN}}$. That is, we train $d_{\text{WILT}}$ to mimic the distances between the graph embeddings of a given MPNN, as shown in Algorithm 1. Once we have trained $w$ by minimizing $\mathcal{L}$, we can gain insight into $d_{\text{MPNN}}$ via $d_{\text{WILT}}$. WL colors with large edge weights are those whose presence in a graph significantly affects $d_{\text{MPNN}}$ between the graph and other graphs. Specifically, we can derive the following reasoning.

    Large difference between $G$ and $H$ in the number or ratio of WL colors with a large $w(e_{\{c,p(c)\}})$

$\Longrightarrow$ Large $d_{\text{WILT}}(G, H)$    ($\because$ Proposition 1)
$\Longrightarrow$ Large $d_{\text{MPNN}}(G, H)$    ($\because d_{\text{WILT}}$ approximates $d_{\text{MPNN}}$)

## 4.5 Expressiveness of Pseudometrics on WILT

Here, we discuss which of the two normalizations is preferred for a given MPNN based on the expressive power. Below are the relationships between the expressiveness of $d_{\text{MPNN}}$ and $d_{\text{WILT}}$.

**Theorem 1** (Expressive Power of the Pseudometrics on WILT). *Let $d_{MPNN}^{mean}$ and $d_{MPNN}^{sum}$ be $d_{MPNN}$ of MPNNs with mean/sum graph poolings, respectively. We also define a pseudometric based on the $L$-iteration WL test:*

$$d_{WL}(G, H) \coloneqq \mathbb{1}_{\{\!\{c_v^{(L)} | v \in V_G\}\!\} \neq \{\!\{c_v^{(L)} | v \in V_H\}\!\}}.$$

*Then, the following inequalities hold for WILT with positive edge weights.*

$$\dot{d}_{WILT} < \bar{d}_{WILT} \cong d_{WL}, \quad d_{MPNN}^{mean} \leq \dot{d}_{WILT}(< \bar{d}_{WILT}), \quad d_{MPNN}^{sum} \leq \bar{d}_{WILT}, \quad d_{MPNN}^{sum} \lesssim \dot{d}_{WILT}.$$

*Proof.* See Appendix B.4. □

Since $\bar{d}_{\text{WILT}}$ is more expressive than $\dot{d}_{\text{WILT}}$, you might think that $\bar{d}_{\text{WILT}}$ is always preferable to approximating $d_{\text{MPNN}}$. However, $\dot{d}_{\text{WILT}}$ is expected to be better at approximating $d_{\text{MPNN}}^{\text{mean}}$, since it provides a tighter bound. Intuitively, this follows from the fact that mean pooling and the size normalization are essentially the same procedure: They both ignore the number of nodes. In contrast, $\bar{d}_{\text{WILT}}$ is expected to work well on $d_{\text{MPNN}}^{\text{sum}}$, which retains the information about the number of nodes and thus cannot be bounded by $\dot{d}_{\text{WILT}}$. We will experimentally confirm these analyses in Section 5. Note that Theorem 1 considers only the binary expressiveness of pseudometrics. Regarding the size of the family of functions that each pseudometric can represent, $d_{\text{MPNN}}$ is expected to be superior to $d_{\text{WILT}}$, because $d_{\text{WILT}}$ is restricted to an optimal transport on the tree for faster computation and better interpretability. Still, in Section 5, we empirically show that $d_{\text{WILT}}$ can approximate $d_{\text{MPNN}}$ well.

## 5 EXPERIMENTS

In this section, we confirm that our proposed $d_{\text{WILT}}$ can successfully approximate $d_{\text{MPNN}}$. Then, we show that the distribution of learned edge weights of WILT is skewed towards 0, and a large part of them can be removed with L1 regularization. Finally, we investigate the WL colors that influence $d_{\text{MPNN}}$ most. Due to space limitations, we report results only for a selection of MPNNs and datasets. Code is available online, and experimental settings and additional results are in Appendix F.

We trained 3-layer GCNs with mean or sum pooling on the three datasets with five different seeds. We then distilled each into two WILTs, one with size normalization and one with dummy node normalization. To evaluate how well a distance $d$ approximates $d_{\text{MPNN}}$, we used a variant of RMSE:

$$\text{RMSE}(d_{\text{MPNN}}, d) := \sqrt{\min_{\alpha \in \mathbb{R}} \frac{1}{|\mathcal{D}|^2} \sum_{(G,H) \in \mathcal{D}^2} \left( \tilde{d}_{\text{MPNN}}(G, H) - \alpha \cdot \tilde{d}(G, H) \right)^2},$$

where $\tilde{d}_{\text{WILT}}$ and $\tilde{d}$ means they are normalized to $[0, 1]$. Intuitively, the closer the RMSE is to zero, the better the alignment is, and zero RMSE means perfect alignment. We do not use the correlation coefficient because it can be one even if $d_{\text{MPNN}}$ is not a constant multiple of $d$: it allows non-zero intercept. Note that the minimization over $\alpha$ can be solved analytically. Table 2 shows the RMSE between $d_{\text{MPNN}}$ and $\dot{d}_{\text{WILT}}$ or $\bar{d}_{\text{WILT}}$. We also include results for $d_{\text{WWL}}$ and $d_{\text{WLOA}}$, which are special cases of $\dot{d}_{\text{WILT}}$ and $\bar{d}_{\text{WILT}}$ with fixed edge weights, respectively. It is obvious that $d_{\text{WILT}}$ aligns with $d_{\text{MPNN}}$ much better than $d_{\text{WWL}}$ and $d_{\text{WLOA}}$. Interestingly, $\dot{d}_{\text{WILT}}$ approximates $d_{\text{MPNN}}(\text{mean})$ better, while $\bar{d}_{\text{WILT}}$ approximates $d_{\text{MPNN}}(\text{sum})$ better, except for $d_{\text{MPNN}}(\text{sum})$ trained on Lipophilicity, where their performance is close. This observation is consistent with the theoretical analysis in Section 4.5.

Next, we look into the distribution of the learned edge weights of WILT. Figure 4 (left) shows the histogram of the edge weights of the WILT with dummy node normalization after distillation from a 3-layer GCN with sum pooling trained on Mutagenicity. The distribution is heavily skewed towards zero. This plot, together with Proposition 1, suggests that the relative position of MPNN graph embeddings is determined based on only a small subset of WL colors. To further verify this idea, we added an L1 regularization term to the objective function $\mathcal{L}$ and minimized it so that $w(e_{\{c,p(c)\}})$ would be set to zero for some colors. Figure 4 (center) shows the RMSE between $d_{\text{MPNN}}$ and the resulting $\bar{d}_{\text{WILT}}$, as well as the ratio of non-zero edge weights, under different L1 coefficient $\lambda$. As expected, the larger $\lambda$ is, the more edge weights are set to zero and the larger the RMSE. However, it is worth noting that $\bar{d}_{\text{WILT}}$ is much better aligned with $d_{\text{MPNN}}$ than $d_{\text{WLOA}}$ even when trained with $\lambda = 1.0$ and about 95% of the edge weights are zero. This good approximation with only 5% non-zero edges implies that MPNNs rely on only a few important WL colors to define $d_{\text{MPNN}}$.

Finally, we show the subgraphs corresponding to the colors with the largest weights, thus influencing $d_{\text{MPNN}}$ the most. Again, we only show results for the 3-layer GCN with sum pooling trained on the Mutagenicity dataset. To avoid identifying colors that are too rare, we only consider colors that

Table 2: The mean±std of RMSE($d_{\mathrm{MPNN}}, d$) [$\times 10^{-2}$] over five different seeds. Each column corresponds to a GCN with a given graph pooling method, trained on a given dataset.

| | Mutagenicity | | ENZYMES | | Lipophilicity | |
|---|---|---|---|---|---|---|
| | mean | sum | mean | sum | mean | sum |
| $d_{\mathrm{WWL}}$ | 9.25±0.87 | 12.25±0.54 | 12.18±0.23 | 11.28±0.65 | 10.92±0.42 | 10.83±0.73 |
| $d_{\mathrm{WLOA}}$ | 18.74±3.36 | 5.98±1.60 | 16.79±2.33 | 6.83±0.41 | 13.97±0.97 | 10.00±1.34 |
| $\dot{d}_{\mathrm{WILT}}$ | **1.74 ± 0.52** | 1.22±0.31 | **2.71 ± 0.38** | 9.15±0.47 | **3.11 ± 0.54** | **2.50 ± 0.67** |
| $\bar{d}_{\mathrm{WILT}}$ | 3.34±1.01 | **0.82 ± 0.17** | 4.64±0.67 | **1.43 ± 0.10** | 6.35±1.22 | 2.64±0.74 |

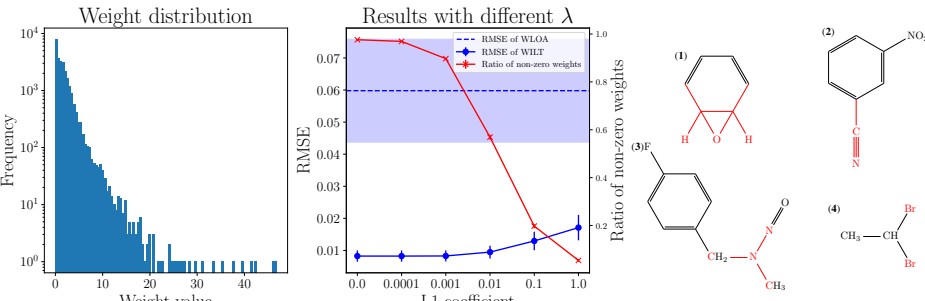

Figure 4: (left): The distribution of the edge weights of WILT after distillation. (center): The RMSE and the ratio of non-zero edge weights after distillation with different coefficients for the L1 term. The results are mean and std over five different seeds. (right): Example graphs with highlighted significant subgraphs corresponding to colors with the largest weights.

appear in at least 1% of the entire graphs. Figure 4 (right) shows example graphs with subgraphs corresponding to colors with the four largest weights. The identified subgraphs in (1) and (4) are known to be characteristic of mutagenic molecules (Kazius et al., 2005). In fact, (1) and (4) are classified as "epoxide" and "aliphatic halide" based on the highlighted subgraphs. Given that only a tiny fraction of the entire WL colors correspond to the subgraphs reported in (Kazius et al., 2005), this result suggests that MPNNs learn the relative position of graph embeddings based on WL colors that are also known to be functionally important by domain knowledge.

## 6 CONCLUSIONS

We analyzed the metric properties of the embedding space of MPNNs. We found that the alignment with the functional pseudometric improves during training and is a key to high predictive performance. In contrast, the alignment with the structural psudometrics, which has been studied intensively in previous works, does not improve and is not correlated with performance. To understand how MPNNs learn and reflect the functional distance between graphs, we propose a theoretically sound and efficiently computable new pseudometric on graphs using WILT. By examining the edge weights of the distilled WILT, we found that only a tiny fraction of the entire WL colors influence $d_{\mathrm{MPNN}}$. The identified colors correspond to subgraphs that are known to be functionally important from domain knowledge.

While we investigated MPNNs specifically, there is a hierarchy of more and more expressive GNNs that are bounded in expressiveness by corresponding WL test variants. In this paper, we have defined WILT on the hierarchy of 1-WL labels. Still, it is straightforward to extend the proposed WILT metric to color hierarchies obtained from higher-order WL variants (Morris et al., 2023; Geerts & Reutter, 2022) or extended message passing schemes (Frasca et al., 2022; Graziani et al., 2024). While beyond the scope of this work, higher-order WILTing trees may prove useful in interpreting a range of GNNs. However, as the number of trainable WILT weights scales with the number of colors, the practical relevance of higher-order WILTs remains an open question. Using WILT for a purpose other than understanding GNNs is also interesting. For example, by training WILT's edge parameters from scratch, we might be able to build a high-performance graph kernel.

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

## A  RELATED WORK

A graph is a data structure composed of nodes and the edges connecting them, capable of representing various entities such as molecules and social networks. Due to the high flexibility of graph structures, it is difficult to apply deep neural networks from non-graph domains, such as convolutional neural networks (Krizhevsky et al., 2012), to graph data. Thus, graph-specific architectures called graph neural networks have been studied for about two decades since their initial proposal (Gori et al., 2005). Message passing graph neural networks, in particular, achieve high predictive performance in various tasks, including node or graph classification and link prediction.

To understand the high performance of MPNNs, many studies have focused on their expressive power (Morris et al., 2019; Xu et al., 2019; Maron et al., 2019). Expressive power refers to the ability of a permutation invariant function to embed nonisomorphic graphs into distinct representations. Formally, a message passing graph embedding function $f$ is said to be at least as expressive as another function $g$ if the following holds:

$$\forall G, H \in \mathcal{G} : f(G) = f(H) \implies g(G) = g(H),$$

where $\mathcal{G}$ is the set of all pairwise nonisomorphic graphs. However, the binary expressive power cannot capture the similarity between graphs, so it alone is said to be insufficient to explain the performance of MPNNs. Recently, it has become increasingly recognized that the geometry of the embedding space in MPNNs, not just their combinatorial expressiveness, is crucial (Li & Leskovec, 2022; Morris et al., 2024). For instance, many of the graph contrastive learning methods implicitly assume that good metric structure in the embedding space will lead to the high performance of MPNNs (Liu et al., 2022b). Chuang & Jegelka (2022) theoretically showed that the distance between MPNN embeddings can be upper bounded by their proposed task-irrelevant structural distance, called the tree mover's distance, paving the way for the theoretical analyses of MPNN generalization ability or robustness. Böker et al. (2024) proved the equivalence between the MPNN embedding distance and other structural distances, but their analyses dealt only with dense graphs and required the consideration of all MPNNs with some Lipschitz constant. Our study also focuses on the geometry of the embedding space, but we investigate one MPNN trained on practical sparse graphs.

This study is also related to GNN interpretability (Liu et al., 2022a; Yuan et al., 2022). The interpretability of GNNs is important because people may be reluctant to apply them to real-world problems where safety or privacy are important if the mechanism behind their predictions is unknown. Furthermore, higher interpretability of well-performing models may lead to a new understanding of scientific phenomena when applied to scientific domains such as chemistry or biology. Most of existing interpretation methods are instance-level, which identify input features in a given input graph that are important for its prediction. However, instance-level methods cannot explain the global behavior of GNNs. Recently, some studies have proposed a way to understand the global behavior of GNNs by distilling them into highly interpretable models. The resulting model can be a GNN with higher interpretability (Müller et al., 2024), or a logical formula (Azzolin et al., 2023; Köhler & Heindorf, 2024; Pluska et al., 2024). Our study also distills MPNNs into highly interpretable WILT for global-level understanding, but the difference is that ours aims to interpret the metric structure of the MPNN embedding space, while previous studies focused on generating explanations for each label class. In addition, our method can be applied to regression tasks on graphs, while previous studies cannot.

## B  THEORETICAL ANALYSIS

### B.1  STRUCTURAL PSEUDOMETRICS

Here we introduce the definitions of the graph edit distance ($d_{\text{GED}}$, Sanfeliu & Fu, 1983), Weisfeiler Leman optimal assignment distance ($d_{\text{WLOA}}$, Kriege et al., 2016), and Wasserstein Weisfeiler Leman graph distance ($d_{\text{WWL}}$, Togninalli et al., 2019). For the definition of tree mover's distance, please refer to the original paper (Chuang & Jegelka, 2022).

**Definition 6** (Graph Edit Distance(Sanfeliu & Fu, 1983))**.** *Let $\mathcal{E}$ be the set of graph edit operations, and $c : \mathcal{E} \to \mathbb{R}_{\geq 0}$ be a function that assigns a cost to each operation. Then, the* graph edit distance

*(GED) between $G$ and $H$ is defined as the minimum cost of a sequence of edit operations that transform $G$ into $H$. Formally,*

$$d_{GED}(G, H) := \min_{s \in S(G,H)} \sum_{e \in s} c(e),$$

*where $S$ is a set of sequences of graph edit operations that transform $G$ into $H$.*

In this paper, $\mathcal{E}$ consists of insertion and deletion of single nodes and single edges, as well as substitution of single node or edge labels. We set the cost of each operation to 1, i.e., $c(e) \equiv 1$. Next, we move on to the Weisfeiler Leman optimal assignment (WLOA) kernel.

**Definition 7** (Weisfeiler Leman Optimal Assignment Kernel (Kriege et al., 2016)). *Consider $G = (V_G, E_G)$ and $H = (V_H, E_H)$. Let $V'_G$ and $V'_H$ be the extended node sets resulting from adding special nodes $z$ to $G$ or $H$ so that $G$ and $H$ have the same number of nodes. Let the base kernel $k$ is defined as:*

$$k(v, u) := \begin{cases} \sum_{l=0}^{L} \mathbb{1}_{c_v^{(l)} = c_u^{(l)}} & (v \neq z \wedge u \neq z) \\ 0 & (v = z \vee u = z), \end{cases}$$

*where $c_v^{(l)}$ and $c_u^{(l)}$ represent the colors of vertices $v$ and $u$ at iteration $l$ of the WL algorithm (see Section 2). Then, the Weisfeiler Leman optimal assignment (WLOA) kernel is defined as:*

$$k_{WLOA}(G, H) := \max_{B \in \mathcal{B}(V'_G, V'_H)} \sum_{(v_G, u_H) \in B} k(v_G, u_H),$$

*where $\mathcal{B}(V'_G, V'_H)$ denotes the set of all possible bijections between $V'_G$ and $V'_H$.*

Kriege et al. (2016) proved that $k_{\text{WLOA}}$ is a positive semidefinite kernel function. While they focus only on the kernel, a corresponding graph pseudometric can be defined in the following way:

**Definition 8** (Weisfeiler Leman Optimal Assignment (WLOA) Distance). *A function $d_{WLOA}$ below is a pseudometric on the set of pairwise nonisomorphic graphs $\mathcal{G}$:*

$$d_{WLOA}(G, H) := (L + 1) \cdot \max(|V_G|, |V_H|) - k_{WLOA}(G, H)$$

*Proof.* Theorem 3 shows that $d_{\text{WLOA}}$ defined as above is a special case of $\bar{d}_{\text{WILT}}$. Since $\bar{d}_{\text{WILT}}$ is a pseudometric on the set of pairwise nonisomorphic graphs $\mathcal{G}$, so is $d_{\text{WLOA}}$. $\qquad\square$

We will show later that the above WLOA distance is a special case of our WILT distance with dummy node normalization (Theorem 3). Togninalli et al. (2019) proposed another graph pseudometric based on the WL algorithm, called Wasserstein Weisfeiler Leman graph distance.

**Definition 9** (Wasserstein Weisfeiler Leman (WWL) Distance (Togninalli et al., 2019)). *Let $d_{ham}(v, u)$ be the hamming distance between $\left[ c_v^{(0)}, c_v^{(1)}, \ldots c_v^{(L)} \right]$ and $\left[ c_u^{(0)}, c_u^{(1)}, \ldots c_u^{(L)} \right]$, where $c_v^{(l)}$ is the color of node $v$ at iteration $l$ of the WL algorithm (see Section 2). Specifically,*

$$d_{ham}(v, u) := \frac{1}{L+1} \sum_{l=0}^{L} \mathbb{1}_{c_v^{(l)} \neq c_u^{(l)}}.$$

*Then the WWL distance is defined as*

$$d_{WWL}(G, H) := \min_{P \in \Gamma_{WWL}} \sum_{v_i \in V_G} \sum_{u_j \in V_H} P_{i,j} d_{ham}(v_i, u_j),$$

*where $\Gamma_{WWL} := \{P \in \mathbb{R}_{\geq 0}^{|V_G| \times |V_H|} \mid P\mathbf{1} = \frac{1}{|V_G|}\mathbf{1}, P^T\mathbf{1} = \frac{1}{|V_H|}\mathbf{1}\}$ is a set of valid transports between two uniform discrete distributions.*

Togninalli et al. (2019) have shown that $d_{\text{WWL}}$ is a pseudometric. In addition, they proposed a corresponding kernel $k_{\text{WWL}}(G, H) := e^{-\lambda d_{\text{WWL}}(G,H)}$, and showed that it is positive semidefinite. We will prove later in Theorem 2 that our WILT distance with size normalization includes the WWL distance as a special case.

## B.2 FUNCTIONAL PSEUDOMETRIC

Here, we show that $d_{\text{func}}$ is a pseudometric.

**Definition 2** (Functional Pseudometric). *Let $y_G$ be the target label of graph $G$ in a given task. In classification, $y_G$ is a categorical class, while $y_G$ is a numerical value in regression. We assume the space for $y_G$ is bounded. Then, the functional pseudometric space $(\mathcal{G}, d_{func})$ is obtained from $d_{func} : \mathcal{G} \times \mathcal{G} \to [0,1]$ defined as:*

$$d_{func}(G, H) := \begin{cases} \mathbb{1}_{y_G \neq y_H} & \text{(classification)} \\ \dfrac{|y_G - y_H|}{\sup\limits_{I \in \mathcal{G}} y_I - \inf\limits_{I \in \mathcal{G}} y_I} & \text{(regression)}, \end{cases}$$

*where $\mathbb{1}_{y_G \neq y_H}$ is the indicator function that returns 1 if $y_G \neq y_H$, otherwise 0.*

*Proof.* We start with the classification case.

$$\begin{aligned} d_{\text{func}}(G, G) &= \mathbb{1}_{y_G \neq y_G} \\ &= 0 \\ d_{\text{func}}(G, H) &= \mathbb{1}_{y_G \neq y_H} \\ &= \mathbb{1}_{y_H \neq y_G} \\ &= d_{\text{func}}(H, G) \\ d_{\text{func}}(G, F) &= \mathbb{1}_{y_G \neq y_F} \\ &\leq \mathbb{1}_{y_G \neq y_H} + \mathbb{1}_{y_H \neq y_F} \\ &= d_{\text{func}}(G, H) + d_{\text{func}}(H, F) \end{aligned}$$

We can prove similarly in regression case.

$$\begin{aligned} d_{\text{func}}(G, G) &= \frac{|y_G - y_G|}{\sup\limits_{I \in \mathcal{G}} y_I - \inf\limits_{I \in \mathcal{G}} y_I} \\ &= 0 \\ d_{\text{func}}(G, H) &= \frac{|y_G - y_H|}{\sup\limits_{I \in \mathcal{G}} y_I - \inf\limits_{I \in \mathcal{G}} y_I} \\ &= \frac{|y_H - y_G|}{\sup\limits_{I \in \mathcal{G}} y_I - \inf\limits_{I \in \mathcal{G}} y_I} \\ &= d_{\text{func}}(H, G) \\ d_{\text{func}}(G, F) &= \frac{|y_G - y_F|}{\sup\limits_{I \in \mathcal{G}} y_I - \inf\limits_{I \in \mathcal{G}} y_I} \\ &\leq \frac{|y_G - y_H|}{\sup\limits_{I \in \mathcal{G}} y_I - \inf\limits_{I \in \mathcal{G}} y_I} + \frac{|y_H - y_F|}{\sup\limits_{I \in \mathcal{G}} y_I - \inf\limits_{I \in \mathcal{G}} y_I} \\ &= d_{\text{func}}(G, H) + d_{\text{func}}(H, F) \end{aligned}$$

In both cases, identity, symmetry, and triangle inequality are satisfied. $\square$

If the sup/inf of $y_G$ in $\mathcal{G}$ are unknown, they can be approximated by the max/min in a training dataset $\mathcal{D}$, and we can similarly prove that $d_{\text{func}}$ is a pseudometric.

## B.3 NORMALIZED WILTING DISTANCES AND RELATIONSHIP TO EXISTING DISTANCES

We present the formal definitions of the size normalization and dummy node normalization. We then show that $\dot{d}_{\text{WILT}}$ with size normalization generalizes the WWL distance and $\bar{d}_{\text{WILT}}$ with dummy node normalization generalizes the WLOA distance.

**Definition 10** (WILTing Distance with Size Normalization). *We define the WILTing distance with size normalization as:*

$$\dot{d}_{WILT}(G, H; w) := \min_{P \in \dot{\Gamma}} \sum_{v_i \in V_G} \sum_{u_j \in V_H} P_{i,j} d_{path}(c_{v_i}^{(L)}, c_{u_j}^{(L)}),$$

*where $\dot{\Gamma} := \{P \in \mathbb{R}^{|V_G| \times |V_H|} \mid P_{i,j} \geq 0, P\mathbf{1} = \frac{1}{|V_G|}\mathbf{1}, P^T\mathbf{1} = \frac{1}{|V_H|}\mathbf{1}\}$. It is equivalent to:*

$$\dot{d}_{WILT}(G, H; w) = \sum_{c \in V(T_{\mathcal{D}}) \setminus \{r\}} w(e_{\{c, p(c)\}}) \left| \dot{\nu}_c^G - \dot{\nu}_c^H \right|,$$

*where $\dot{\nu}^G := \frac{1}{|V_G|} \nu^G$.*

The only difference between Definition 4 and Definition 10 is the mass assigned to each node. The equivalence between the two definitions of $\dot{d}_{\text{WILT}}$ is a straightforward consequence of (Le et al., 2019). The other normalization is defined as follows.

**Definition 11** (WILTing Distance with Dummy Node Normalization). *Let $\bar{V}_G$ be an extension of $V_G$ with additional $N - |V_G|$ isolated dummy nodes with special label, where $N := \max_{G \in \mathcal{D}} |V_G|$. Let $\bar{T}_{\mathcal{D}}$ be WILT built from the extended graphs $\{(\bar{V}_G, E_G)\}_{G \in \mathcal{D}}$. Note that $\bar{T}_{\mathcal{D}}$ is just a slight modification of $T_{\mathcal{D}}$ (see Figure 3). We define the WILTing distance with dummy node normalization as:*

$$\bar{d}_{WILT}(G, H; w) := \min_{P \in \bar{\Gamma}} \sum_{v_i \in \bar{V}_G} \sum_{u_j \in \bar{V}_H} P_{i,j} d_{path}(\bar{c}_{v_i}^{(L)}, \bar{c}_{u_j}^{(L)}),$$

*where $\bar{\Gamma} := \{P \in \mathbb{R}^{|\bar{V}_G| \times |\bar{V}_H|} \mid P_{i,j} \geq 0, P\mathbf{1} = \mathbf{1}, P^T\mathbf{1} = \mathbf{1}\}$, and $\bar{c}_v^{(L)}$ is the color of node $v$ on $\bar{T}_{\mathcal{D}}$ after $L$ iterations. An equivalent definition is:*

$$\bar{d}_{WILT}(G, H; w) = \sum_{\bar{c} \in V(\bar{T}_{\mathcal{D}}) \setminus \{r\}} w(e_{\{\bar{c}, p(\bar{c})\}}) \left| \bar{\nu}_{\bar{c}}^G - \bar{\nu}_{\bar{c}}^H \right|,$$

*where $\bar{\nu}^G$ is the WILT embedding of $G$ using $\bar{T}_{\mathcal{D}}$.*

Intuitively speaking, we add dummy nodes to all the graphs so that they have the same number of nodes[1], and compute the WILTing distance in exactly the same way as shown in Section 4.2.

Next, we show that $\dot{d}_{\text{WILT}}$ includes the Wasserstein Weisfeiler Leman distance and $\bar{d}_{\text{WILT}}$ includes the Weisfeiler Leman optimal assignment distance as a special case, respectively.

**Theorem 2** ($d_{\text{WWL}}$ as a Special Case of $\dot{d}_{\text{WILT}}$). *The WWL distance in Definition 9 is equal to the WILTing distance with size normalization with all WILT edge weights set to $\frac{1}{2(L+1)}$. Specifically,*

$$d_{WWL}(G, H) = \dot{d}_{WILT}\left(G, H; w \equiv \frac{1}{2(L+1)}\right).$$

*Proof.*

$$d_{\text{WWL}}(G, H) := \min_{P \in \Gamma_{\text{WWL}}} \sum_{v_i \in V_G} \sum_{u_j \in V_H} P_{i,j} d_{\text{ham}}(v_i, u_j)$$

$$= \min_{P \in \Gamma_{\text{WWL}}} \sum_{v_i \in V_G} \sum_{u_j \in V_H} P_{i,j} \frac{1}{L+1} \sum_{l=0}^{L} \mathbb{1}_{c_v^l \neq c_u^l}$$

$$= \min_{P \in \Gamma_{\text{WWL}}} \sum_{v_i \in V_G} \sum_{u_j \in V_H} P_{i,j} d_{\text{path}}\left(c_{v_i}^{(L)}, c_{u_j}^{(L)}; w \equiv \frac{1}{2(L+1)}\right)$$

$$= \min_{P \in \dot{\Gamma}} \sum_{v_i \in V_G} \sum_{u_j \in V_H} P_{i,j} d_{\text{path}}\left(c_{v_i}^{(L)}, c_{u_j}^{(L)}; w \equiv \frac{1}{2(L+1)}\right)$$

$$= \dot{d}_{\text{WILT}}\left(G, H; w \equiv \frac{1}{2(L+1)}\right).$$

$\square$

---

[1] In fact, $\bar{d}_{\text{WILT}}$ remains a pseudometric even on $\mathcal{D} = \mathcal{G}$, as it can be defined without explicit use of $N$. To this end, note that $\lim_{N \to \infty} |\bar{\nu}_{c_{\urcorner}^i}^G - \bar{\nu}_{c_{\urcorner}^i}^H| = |V(G) - V(H)|$ for any dummy node color $c_{\urcorner}^i$.

**Theorem 3** ($d_{\mathrm{WLOA}}$ as a Special Case of $\bar{d}_{\mathrm{WILT}}$). *The WLOA distance in Definition 8 is equal to the WILTing distance with dummy node normalization with all WILT edge weights set to $\frac{1}{2}$. Specifically,*

$$d_{WLOA}(G, H) = \bar{d}_{WILT}\left(G, H; w \equiv \frac{1}{2}\right).$$

*Proof.* First, $d_{\mathrm{WLOA}}(G, H)$ can be transformed as follows.

$$\begin{aligned}
d_{\mathrm{WLOA}}(G, H) &:= (L+1) \cdot \max(|V_G|, |V_H|) - k_{\mathrm{WLOA}}(G, H) \\
&= (L+1) \cdot \max(|V_G|, |V_H|) - \max_{B \in \mathcal{B}(V_G', V_H')} \sum_{(v_G, u_H) \in B} k(v_G, u_H) \\
&= \min_{B \in \mathcal{B}(V_G', V_H')} \sum_{(v_G, u_H) \in B} (L + 1 - k(v_G, u_H))
\end{aligned}$$

Next, we introduce a equivalent definition of $k(v, u)$. In Definition 7, the WL algorithm is applied only on $V_G$ and $V_H$, not on special nodes. Assume w.l.o.g. that $|V(G)| \leq |V(H)|$, i.e., $V(G)$ is extended with $|V(H)| - |V(G)|$ dummy nodes. By treating the special nodes in $V_G'$ as dummy nodes, we can define WL colors for the special nodes $z$: $(c_z^{(0)}, c_z^{(1)}, \ldots, c_z^{(L)}) = (c_\lrcorner^0, c_\lrcorner^1, \ldots, c_\lrcorner^L)$. Then, as only $V_G'$ contains special nodes, $k(v, u)$ can be simplified to:

$$k(v, u) = \sum_{l=0}^{L} \mathbb{1}_{\bar{c}_v^{(l)} = \bar{c}_u^{(l)}},$$

where $\bar{c}_v^{(l)}$ is the color of node $v$ on the WILT $\bar{T}_{\mathcal{D}}$ with dummy node normalization after $l$ iterations. Then, $L + 1 - k(v, u)$ is equivalent to $d_{\mathrm{path}}(\bar{c}_v^{(L)}, \bar{c}_u^{(L)}; w \equiv \frac{1}{2})$:

$$\begin{aligned}
L + 1 - k(v, u) &= L + 1 - \sum_{l=0}^{L} \mathbb{1}_{\bar{c}_v^{(l)} = \bar{c}_u^{(l)}} \\
&= \sum_{l=0}^{L} \mathbb{1}_{\bar{c}_v^{(l)} \neq \bar{c}_u^{(l)}} \\
&= d_{\mathrm{path}}\left(\bar{c}_v^{(L)}, \bar{c}_u^{(L)}; w \equiv \frac{1}{2}\right)
\end{aligned}$$

Therefore, $d_{\mathrm{WLOA}}$ is a special case of $\bar{d}_{\mathrm{WILT}}$:

$$\begin{aligned}
d_{\mathrm{WLOA}}(G, H) &= \min_{B \in \mathcal{B}(V_G', V_H')} \sum_{(v_G, u_H) \in B} (L + 1 - k(v_G, u_H)) \\
&= \min_{B \in \mathcal{B}(V_G', V_H')} \sum_{(v_G, u_H) \in B} d_{\mathrm{path}}\left(\bar{c}_{v_G}^{(L)}, \bar{c}_{u_H}^{(L)}; w \equiv \frac{1}{2}\right) \\
&\stackrel{\star}{=} \min_{P \in \bar{\Gamma}} \sum_{v_i \in \bar{V}_G} \sum_{u_j \in \bar{V}_H} P_{i,j} d_{\mathrm{path}}\left(\bar{c}_{v_i}^{(L)}, \bar{c}_{u_j}^{(L)}; w \equiv \frac{1}{2}\right) \\
&= \bar{d}_{\mathrm{WILT}}\left(G, H; w \equiv \frac{1}{2}\right)
\end{aligned}$$

Note that $\star$ holds since adding the same number of dummy nodes to both $G$ and $H$ does not change the left side, and the optimal transport on WILT always delivers a mass on a node to only one node. $\square$

## B.4 EXPRESSIVENESS OF GRAPH PSEUDOMETRICS

We now discuss in detail the expressiveness of graph pseudometrics, which was summarized in Section 4.5. We split Theorem 1 in Section 4.5 into three theorems below, and prove each one

separately. The discussion below provides a possible explanation for some results in Section 5 and Appendix E. First, we introduce a pseudometric defined by the WL test:

$$d_{\text{WL}}(G, H) := \mathbb{1}_{\{\!\!\{c_v^{(L)} \,|\, v \in V_G\}\!\!\} = \{\!\!\{c_v^{(L)} \,|\, v \in V_H\}\!\!\}},$$

where $L$ is the number of WL iterations. In other words, $d_{\text{WL}}(G, H) = 1$ if the $L$-iteration WL test can distinguish $G$ and $H$, otherwise 0. With this definition, we start with the comparison of $d_{\text{WILT}}$ and $d_{\text{WL}}$ for a better understanding of $d_{\text{WILT}}$.

**Theorem 4** (Expressiveness of the WILTing Distance). *Suppose $\dot{d}_{WILT}$ and $\bar{d}_{WILT}$ are pseudometrics defined with WILT with some edge weight functions. We assume that all edge weights are positive for $\bar{d}_{WILT}$. Then,*

$$\dot{d}_{WILT} < \bar{d}_{WILT} \cong d_{WL}.$$

*Proof.* We first show $\dot{d}_{\text{WILT}} \leq \bar{d}_{\text{WILT}}$.

$$
\begin{aligned}
\bar{d}_{\text{WILT}}(G, H) = 0 &\implies \bar{\nu}^G = \bar{\nu}^H \quad \wedge \quad |V_G| = |V_H| \\
&\implies \forall \text{ leaf color } c : \quad |\{v \in V_G \mid c_v^{(L)} = c\}| = |\{v \in V_H \mid c_v^{(L)} = c\}| \\
&\implies \forall \text{ leaf color } c : \quad \frac{|\{v \in V_G \mid c_v^{(L)} = c\}|}{|\{v \in V_H \mid c_v^{(L)} = c\}|} = \frac{|V_G|}{|V_H|} = 1 \\
&\implies \dot{\nu}^G = \dot{\nu}^H \\
&\implies \dot{d}_{\text{WILT}}(G, H) = 0.
\end{aligned}
$$

Note that leaf color $c$ means that $c$ is a leaf of the WILT. The first implication follows from the fact that dummy node normalization implies that only graphs with identical numbers of nodes can have a distance of zero if the weights are positive. To see that $\bar{d}_{\text{WILT}}(G, H)$ is more expressive than $\dot{d}_{\text{WILT}}(G, H)$, note that there are $G$ and $H$ s.t. $\bar{d}_{\text{WILT}}(G, H) \neq 0 \wedge \dot{d}_{\text{WILT}}(G, H) = 0$: For example, let $G$ and $H$ be $k$-regular graphs (such as cycles) with different numbers of nodes and identical node and edge labels. Next, we show $\bar{d}_{\text{WILT}} = d_{\text{WL}}$.

$$
\begin{aligned}
\bar{d}_{\text{WILT}}(G, H) = 0 &\iff \bar{\nu}^G = \bar{\nu}^H \\
&\iff \forall \text{ leaf color } c : \quad |\{v \in V_G \mid c_v^{(L)} = c\}| = |\{v \in V_H \mid c_v^{(L)} = c\}| \\
&\iff \{\!\!\{c_v^{(L)} \mid v \in V_G\}\!\!\} = \{\!\!\{c_v^{(L)} \mid v \in V_H\}\!\!\} \\
&\iff d_{\text{WL}}(G, H) = 0.
\end{aligned}
$$

The first equivalence again follows from the fact that weights are positive. $\qquad\square$

Since $d_{\text{MPNN}} \leq d_{\text{WL}}$ holds for any MPNN (Xu et al., 2019), the above theorem implies that $d_{\text{MPNN}} \leq \bar{d}_{\text{WILT}}$ if all edge weights are positive. At first glance, this seems to suggest that $\bar{d}_{\text{WILT}}$ can better align with any MPNN than $\dot{d}_{\text{WILT}}$ because of its high expressiveness. However, the results in Section 5 show that $\dot{d}_{\text{WILT}}$ is suitable for MPNNs with mean pooling, while $\bar{d}_{\text{WILT}}$ is suitable for MPNNs with sum pooling. Next, we compare $d_{\text{MPNN}}$ and $d_{\text{WILT}}$ in more detail to interpret these results. We start with MPNNs with mean pooling, whose pseudometrics we will call $d_{\text{MPNN}}^{\text{mean}}$.

**Theorem 5** (Expressiveness of the Pseudometric of MPNN with Mean Pooling). *Suppose $\dot{d}_{WILT}$ and $\bar{d}_{WILT}$ are pseudometrics defined with WILT with some edge weight functions. We assume that all edge weights are positive. Then,*

$$d_{MPNN}^{mean} \leq \dot{d}_{WILT}(< \bar{d}_{WILT}).$$

*Proof.* We first show the left inequality.

$$\dot{d}_{\text{WILT}}(G, H) = 0 \implies \dot{\nu}^G = \dot{\nu}^H$$

$$\implies \forall \text{ leaf color } c: \quad \frac{|\{v \in V_G \mid c_v^{(L)} = c\}|}{|V_G|} = \frac{|\{v \in V_H \mid c_v^{(L)} = c\}|}{|V_H|}$$

$$\implies \forall \text{ leaf color } c: \quad \frac{1}{|V_G|} \sum_{v \in V_G : c_v^{(L)} = c} h_v^{(L)} = \frac{1}{|V_H|} \sum_{v \in V_H : c_v^{(L)} = c} h_v^{(L)}$$

$$\implies \frac{1}{|V_G|} \sum_{v \in V_G} h_v^{(L)} = \frac{1}{|V_H|} \sum_{v \in V_H} h_v^{(L)}$$

$$\implies d_{\text{MPNN}}^{\text{mean}}(G, H) = 0.$$

The first implication follows from the fact that $w(e_{\{c,p(c)\}}) > 0$ for all colors. The third implication follows from Xu et al. (2019) by noting that $c_u^{(L)} = c_v^{(L)} \implies h_u^{(L)} = h_v^{(L)}$ for any MPNN.

$\dot{d}_{\text{WILT}} < \bar{d}_{\text{WILT}}$ follows from Theorem 4. $\qquad\square$

In Section 5, we found that RMSE($d_{\text{MPNN}}^{\text{mean}}$, $\dot{d}_{\text{WILT}}$) is smaller than RMSE($d_{\text{MPNN}}^{\text{mean}}$, $\bar{d}_{\text{WILT}}$). The above theorem and the proof yield an interpretation of the result. In terms of expressiveness, $\dot{d}_{\text{WILT}}$ is a stricter upper bound on $d_{\text{MPNN}}^{\text{mean}}$ than $\bar{d}_{\text{WILT}}$, since the mean pooling and the size normalization are essentially the same procedure. Both ignore the information about the number of nodes in graphs. When we try to fit $\bar{d}_{\text{WILT}}$ to $d_{\text{MPNN}}^{\text{mean}}$, it is difficult to tune edge parameters so that $\bar{d}_{\text{WILT}}$ can ignore the number of nodes in graphs, but $\dot{d}_{\text{WILT}}$ satisfies this property by definition. This may be the reason why $\dot{d}_{\text{WILT}}$ can be trained to be better aligned with $d_{\text{MPNN}}^{\text{mean}}$ than $\bar{d}_{\text{WILT}}$. A similar discussion can be applied to $d_{\text{WWL}}$ and $d_{\text{WLOA}}$, which are special cases of $\dot{d}_{\text{WILT}}$ and $\bar{d}_{\text{WILT}}$, respectively. Next, we analyze MPNNs with sum pooling.

**Theorem 6** (Expressiveness of the Pseudometric of MPNN with Sum Pooling). *Suppose $\bar{d}_{WILT}$ is defined with WILT with an edge weight function that assigns a positive value to all edges. Then,*

$$d_{MPNN}^{sum} \leq \bar{d}_{WILT}.$$

*In addition, if $\exists G \in \mathcal{G}$ s.t. $\sum_{v \in V_G} h_v^{(L)} \neq 0$, then*

$$d_{MPNN}^{sum} \lneq \dot{d}_{WILT}$$

*Proof.* We begin with $d_{\text{MPNN}}^{\text{sum}} \leq \bar{d}_{\text{WILT}}$.

$$\bar{d}_{\text{WILT}}(G, H) = 0 \implies \bar{\nu}^G = \bar{\nu}^H$$

$$\implies \forall \text{ leaf color } c: \quad |\{v \in V_G \mid c_v^{(L)} = c\}| = |\{v \in V_H \mid c_v^{(L)} = c\}|$$

$$\implies \forall \text{ leaf color } c: \quad \sum_{v \in V_G : c_v^{(L)} = c} h_v^{(L)} = \sum_{v \in V_H : c_v^{(L)} = c} h_v^{(L)}$$

$$\implies \sum_{v \in V_G} h_v^{(L)} = \sum_{v \in V_H} h_v^{(L)}$$

$$\implies d_{\text{MPNN}}^{\text{sum}}(G, H) = 0.$$

Next, we show $d_{\text{MPNN}}^{\text{sum}} \lneq \dot{d}_{\text{WILT}}$. Let $G$ be a graph that satisfies $\sum_{v \in V_G} h_v^{(L)} \neq 0$. We can consider a graph $H$ that consists of two copies of $G$. Then, $\dot{\nu}^G = \dot{\nu}^H$, since $2\nu^G = \nu^H$ and $2|V_G| = |V_H|$.

---

**Algorithm 2** Building WILT

---

**Input**: Graph dataset $\mathcal{D}$
**Parameter**: $L \geq 1$
**Output**: WILT $T_\mathcal{D}$

  $T_\mathcal{D} \leftarrow$ Initial tree with only the root $r$
  **for** $G$ in $\mathcal{D}$ **do**
    /* $L$-iteration WL test on $G$ */
    $c_\mathrm{pre} \leftarrow []$                                  $\triangleright$ Keeping colors in the previous iteration
    $c_\mathrm{now} \leftarrow []$                                  $\triangleright$ Keeping colors in the current iteration
    **for** $v$ in $V_G$ **do**
      **if** $l_\mathrm{node}(v) \notin V(T_\mathcal{D})$ **then**
        $V(T_\mathcal{D}) \leftarrow V(T_\mathcal{D}) \cup \{l_\mathrm{node}(v)\}$
        $E(T_\mathcal{D}) \leftarrow E(T_\mathcal{D}) \cup \{(r, l_\mathrm{node}(v))\}$
      **end if**
      $c_\mathrm{pre}[v] \leftarrow l_\mathrm{node}(v)$
    **end for**
    **for** $l = 1$ to $L$ **do**
      **for** $v$ in $V_G$ **do**
        $c_v \leftarrow \mathrm{HASH}((c_\mathrm{pre}[v], \{\!\!\{(c_\mathrm{pre}[u], l_\mathrm{edge}(e_{uv})) \mid u \in \mathcal{N}(v)\}\!\!\}))$
        **if** $c_v \notin V(T_\mathcal{D})$ **then**
          $V(T_\mathcal{D}) \leftarrow V(T_\mathcal{D}) \cup \{c_v\}$
          $E(T_\mathcal{D}) \leftarrow E(T_\mathcal{D}) \cup \{(c_\mathrm{pre}[v], c_v)\}$
        **end if**
        $c_\mathrm{now}[v] \leftarrow c_v$
      **end for**
      $c_\mathrm{pre} \leftarrow c_\mathrm{now}$
      $c_\mathrm{now} \leftarrow []$
    **end for**
  **end for**
  **return** $T_\mathcal{D}$

---

Therefore, $\dot{d}_\mathrm{WILT}(G, H) = 0$. On the other hand,

$$
\begin{aligned}
d_\mathrm{MPNN}^\mathrm{sum}(G, H) &= \left\| \sum_{v \in V_G} h_v^{(L)} - \sum_{v \in V_H} h_v^{(L)} \right\|_2 \\
&= \left\| \sum_{v \in V_G} h_v^{(L)} - 2 \sum_{v \in V_G} h_v^{(L)} \right\|_2 \\
&= \left\| \sum_{v \in V_G} h_v^{(L)} \right\|_2 \\
&\neq 0.
\end{aligned}
$$

$\square$

In terms of expressiveness, $d_\mathrm{MPNN}^\mathrm{sum}$ is almost always not bounded by $\dot{d}_\mathrm{WILT}$ except for the trivial MPNN which embeds all graphs to zero. In fact, the opposite $\dot{d}_\mathrm{WILT} \leq d_\mathrm{MPNN}^\mathrm{sum}$ holds if the MPNN is sufficiently expressive, e.g. GIN. These analyses may explain why $\mathrm{RMSE}(d_\mathrm{MPNN}^\mathrm{sum}, \bar{d}_\mathrm{WILT})$ is generally smaller than $\mathrm{RMSE}(d_\mathrm{MPNN}^\mathrm{sum}, \dot{d}_\mathrm{WILT})$ in Section 5. No matter how much it is trained, $\dot{d}_\mathrm{WILT}$ cannot capture the information about the number of nodes that $d_\mathrm{MPNN}^\mathrm{sum}$ can. On the other hand, $\bar{d}_\mathrm{WILT}$ is expressive enough to capture the information, and thus has a chance of aligning well with $d_\mathrm{MPNN}^\mathrm{sum}$. Again, a similar reasoning can be applied to $d_\mathrm{WWL}$ and $d_\mathrm{WLOA}$.

## C  ALGORITHM TO CONSTRUCT WILT

Algorithm 2 shows how to build WILT from a graph dataset $\mathcal{D}$.

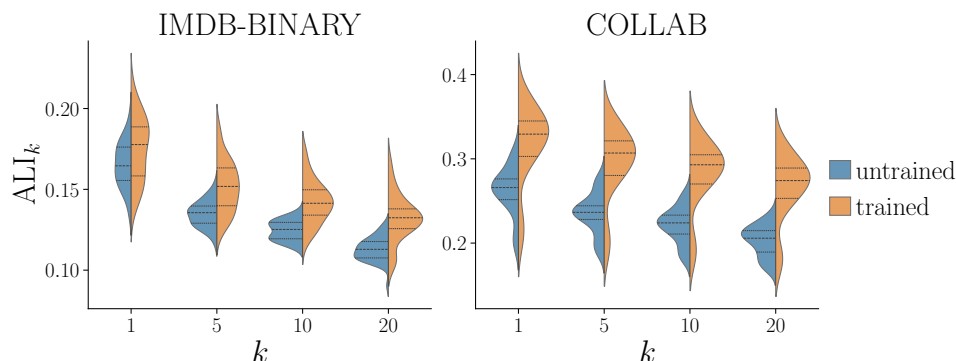

Figure 5: The distribution of $\text{ALI}_k(d_{\text{MPNN}}, d_{\text{func}})$ under different $k$ and datasets.

Table 3: Correlation between $\text{ALI}_k(d_{\text{MPNN}}, d_{\text{func}})$ and accuracy on $\mathcal{D}_{\text{train}}$ and $\mathcal{D}_{\text{test}}$ under different $k$.

|  | IMDB-BINARY | | | | COLLAB | | | |
|---|---|---|---|---|---|---|---|---|
| k | 1 | 5 | 10 | 20 | 1 | 5 | 10 | 20 |
| train | 0.38 | 0.60 | 0.60 | 0.59 | 0.89 | 0.90 | 0.88 | 0.89 |
| test | -0.03 | 0.14 | 0.19 | 0.28 | 0.81 | 0.83 | 0.81 | 0.81 |

## D  EXPERIMENTAL DETAILS FOR SECTION 3

Here, we present the detailed experimental setup resulting in Figure 2 and Table 1. We conduct experiments on three different datasets: Mutagenicity and ENZYMES (Morris et al., 2020), and Lipophilicity (Wu et al., 2018). We chose these datasets to represent binary classification, multiclass classification, and regression tasks, respectively. For the models, we adopt two popular MPNN architectures: GCN and GIN. For each model architecture, we vary the number of message passing layers $(1, 2, 3, 4)$, the embedding dimensions $(32, 64, 128)$, and the graph pooling methods (mean, sum). This results in a total of $2 \times 4 \times 3 \times 2 = 48$ different MPNNs for each dataset. In each setting, we split the dataset into $\mathcal{D}_{\text{train}}$, $\mathcal{D}_{\text{eval}}$, and $\mathcal{D}_{\text{test}}$ (8:1:1). We train the model for 100 epochs and record the performance on $\mathcal{D}_{\text{eval}}$ after each epoch. We set the batch size to 32, and use the Adam optimizer with learning rate of $10^{-3}$. $\text{ALI}_k(d_{\text{MPNN}}, d_{\text{func}})$ and the performance metric (accuracy for Mutagenicity and ENZYMES, RMSE for Lipophilicity) are calculated with the model at the epoch that performed best on $\mathcal{D}_{\text{eval}}$.

Next, we offer additional experimental results on non-molecular datasets: IMDB-BINARY and COLLAB (Morris et al., 2020). Figure 5 visualizes the distribution of $\text{ALI}_k(d_{\text{MPNN}}, d_{\text{func}})$ on these datasets and varying $k$. Similar to Figure 2, $\text{ALI}_k$ consistently improves with training. Table 3 also offers results similar to Table 1, showing that there is a positive correlation between $\text{ALI}_k$ of trained MPNNs and their accuracy in general. We visualize in Figure 6 the plots used to compute the correlation coefficient in Table 1 and Table 3 for better understanding. Each blue dot represents one of the 48 different models. For $\text{ALI}_k$ with $k \neq 5$, similar plots were observed.

## E  MPNN PSEUDOMETRIC AND STRUCTURAL PSEUDOMETRICS

There has been intensive research on graph kernels, which essentially aims to manually design graph pseudometrics $d_{\text{struc}}$ that lead to good prediction performance. Recent studies have theoretically analyzed the relationship between $d_{\text{MPNN}}$ and such $d_{\text{struc}}$, but they only upper-bounded $d_{\text{MPNN}}$ with $d_{\text{struc}}$ (Chuang & Jegelka, 2022), or showed the equivalence for untrained MPNNs on dense graphs (Böker et al., 2024). Therefore, this section examines if $d_{\text{MPNN}}$ really aligns with $d_{\text{struc}}$ in practice, and if the alignment explain the high performance of MPNNs. Specifically, we address the following questions:

**Q1.3** What kind of $d_{\text{struc}}$ is $d_{\text{MPNN}}$ best aligned with?

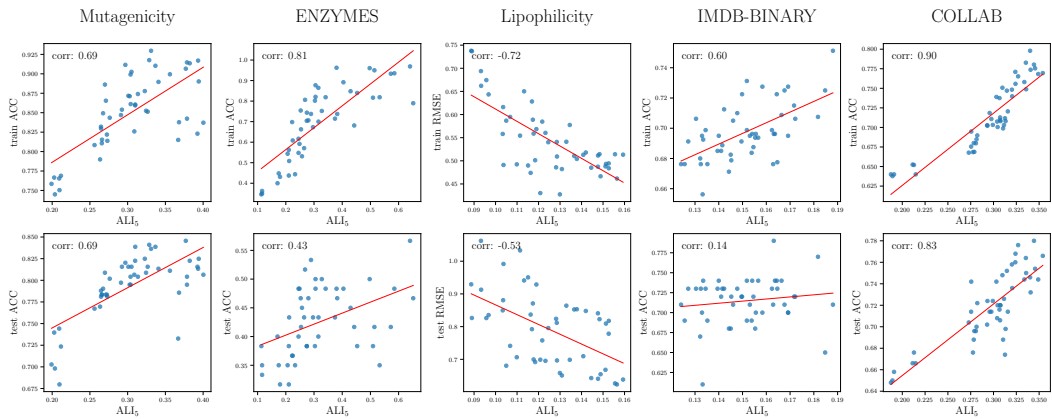

Figure 6: Scatter plots between $\text{ALI}_5(d_{\text{MPNN}}, d_{\text{func}})$ and the performance on the train/test set. In general, higher $\text{ALI}_5(d_{\text{MPNN}}, d_{\text{func}})$, i.e., higher alignment between $d_{\text{MPNN}}$ and $d_{\text{func}}$, indicates higher performance.

**Q1.4** Does training MPNN increase the alignment?

**Q1.5** Does a strong alignment between $d_{\text{MPNN}}$ and $d_{\text{struc}}$ indicate high performance of the MPNN?

We first define an evaluation criterion for the alignment between $d_{\text{MPNN}}$ and $d_{\text{struc}}$ to answer them, which is the same as the one used in Section 5.

**Definition 12** (Evaluation Criterion for Alignment Between $d_{\text{MPNN}}$ and $d_{\text{struc}}$). *Consider a graph dataset denoted by $\mathcal{D}$. Let $\tilde{d}_{\text{MPNN}}$ and $\tilde{d}_{\text{struc}}$ be normalized versions of $d_{\text{MPNN}}$ and $d_{\text{struc}}$, respectively:*

$$\tilde{d}_{\text{MPNN}}(G, H) \coloneqq \frac{d_{\text{MPNN}}(G, H)}{\max\limits_{(G', H') \in \mathcal{D}^2} d_{\text{MPNN}}(G', H')}, \quad \tilde{d}_{\text{struc}}(G, H) \coloneqq \frac{d_{\text{struc}}(G, H)}{\max\limits_{(G', H') \in \mathcal{D}^2} d_{\text{struc}}(G', H')}.$$

*We measure the alignment between $d_{\text{MPNN}}$ and $d_{\text{struc}}$ by the RMSE after fitting a linear model with the intercept fixed at zero to the normalized pseudometrics:*

$$RMSE(d_{\text{MPNN}}, d_{\text{struc}}) \coloneqq \sqrt{\min_{\alpha \in \mathbb{R}} \frac{1}{|\mathcal{D}|^2} \sum_{(G,H) \in \mathcal{D}^2} \left( \tilde{d}_{\text{MPNN}}(G, H) - \alpha \cdot \tilde{d}_{\text{struc}}(G, H) \right)^2}.$$

The closer the RMSE is to zero, the better the alignment is. Zero RMSE means perfect alignment. That is, $d_{\text{MPNN}}$ is a constant multiple of $d_{\text{struc}}$. Note that we use different evaluation criteria to measure the alignment between $d_{\text{MPNN}}$ and $d_{\text{func}}$ (Definition 3) or $d_{\text{struc}}$ (Definition 12). There are multiple reasons for this. First, the RMSE is in principle designed for non-binary $d_{\text{struc}}$. Therefore, $RMSE(d_{\text{MPNN}}, d_{\text{func}})$ is not a meaningful value when $d_{\text{func}}$ is a binary function, which is the case when the task is classification. Second, the computation of $\text{ALI}_k(d_{\text{MPNN}}, d_{\text{struc}})$ is computationally too expensive. We explain this in terms of how many graph pairs we need to compute the distance for. Both the RMSE and $\text{ALI}_k$ require the calculation of the distance between $|\mathcal{D}|^2$ pairs in the original definition. This is too demanding, especially when $d_{\text{struc}}$ is $d_{\text{GED}}$, which is NP-hard to compute. Therefore, in practice, we approximate the RMSE with 1000 randomly selected pairs from $\mathcal{D}^2$. This kind of approximation is difficult for $\text{ALI}_k$. To approximate $\text{ALI}_k$, we first choose a subset $\mathcal{D}_{\text{sub}}$ of $\mathcal{D}$, and then compute $d_{\text{struc}}$ of all pairs in $\mathcal{D}_{\text{sub}}^2$. Even if we set $|\mathcal{D}_{\text{sub}}| = 100$, which is quite small, we still need about 10 times more computation than the RMSE.

We evaluate four structural pseudometrics: graph edit distance ($d_{\text{GED}}$, Sanfeliu & Fu, 1983), tree mover's distance ($d_{\text{TMD}}$, Chuang & Jegelka, 2022), Weisfeiler Leman optimal assignment distance ($d_{\text{WLOA}}$, Kriege et al., 2016), and Wasserstein Weisfeiler Leman graph distance ($d_{\text{WWL}}$, Togninalli et al., 2019). See Appendix B.1 for detailed definitions. $d_{\text{TMD}}$, $d_{\text{WLOA}}$, and $d_{\text{WWL}}$ are pseudometrics on the set of pairwise nonisomorphic graphs $\mathcal{G}$. Only $d_{\text{GED}}$ for strictly positive edit costs is a metric, i.e., $d_{\text{GED}}(G, H) = 0$ if and only if $G$ and $H$ are isomorphic. We will also call $d_{\text{GED}}$ a pseudometric

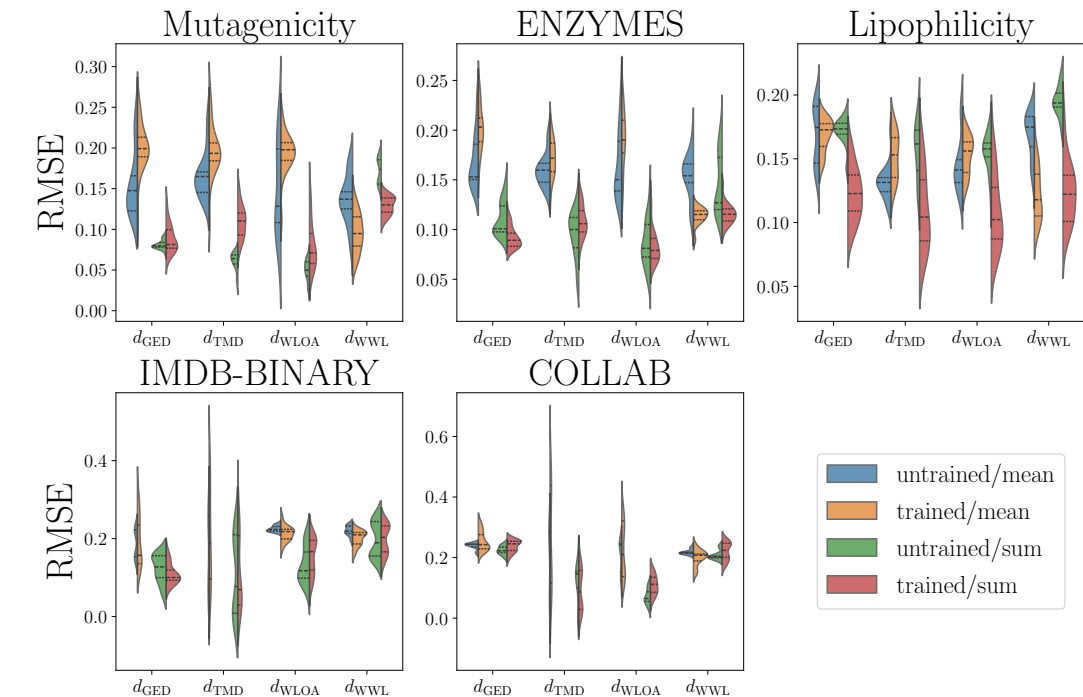

Figure 7: The distributions of $\text{RMSE}(d_{\text{MPNN}}, d_{\text{struc}})$ under different $d_{\text{struc}}$ and datasets. Each color represents whether the MPNNs are trained or not and which graph pooling function they use.

for simplicity. We chose $d_{\text{GED}}$ because it is a popular graph pseudometric. The others were chosen because they are based on the message passing algorithm, like MPNNs, and classifiers based on their corresponding kernels were reported to achieve high accuracy. In addition, $d_{\text{TMD}}$ has been theoretically proven to be an upper bound of $d_{\text{MPNN}}$ (Chuang & Jegelka, 2022). Note that the exact calculation of $d_{\text{GED}}$ is in general NP-hard due to the combinatorial optimization over the set of valid transformation sequences (see Definition 6). Therefore, in our experiment, we limit the computation time of $d_{\text{GED}}$ of each graph pair $(G, H)$ to a maximum of 30 seconds. If this time limit is exceeded, we consider the lowest total cost at that point to be $d_{\text{GED}}(G, H)$. When we compute the RMSE between a given MPNN and any of $d_{\text{TMD}}$, $d_{\text{WLOA}}$, and $d_{\text{WWL}}$, we set the depth of the computational trees used to compute these $d_{\text{struc}}$ as the number of message passing layers in the MPNN for a fair comparison.

Figure 7 presents the distributions of the RMSE in different datasets (Morris et al., 2020; Wu et al., 2018), $d_{\text{struc}}$, and the graph grouping methods used in MPNN. We followed exactly the same procedure for training and evaluating MPNNs as shown in Appendix D. Each distribution consists of $\text{RMSE}(d_{\text{MPNN}}, d_{\text{struc}})$ of 24 MPNNs with different architectures and hyperparameters. We also provide results for untrained MPNNs to see the effect of training. As can be seen from the plots, the distributions of the untrained and trained MPNNs overlap, and there is no strong and consistent improvement in RMSE after training (answer to **Q1.4**). Regarding **Q1.3**, none of the four $d_{\text{struc}}$ performs best in all cases. The best one depends on the choice of dataset and pooling. One intersting observation is that $d_{\text{MPNN}}$ with sum pooling is more aligned with $d_{\text{WLOA}}$ than $d_{\text{WWL}}$, while the reverse is true for $d_{\text{MPNN}}$ with mean pooling. This difference between pooling methods can be explained by different normalizations of the structural pseudometrics (see Section 4.5 and Appendix B.4).

Another insight from Figure 7 is that the degree of alignment between $d_{\text{MPNN}}$ and $d_{\text{struc}}$ varies by model. To see if the alignment is crucial for the high predictive performance of MPNNs, we examined the PCC between $\text{RMSE}(d_{\text{MPNN}}, d_{\text{struc}})$ of trained models and their performance on the training and test sets. We used accuracy and RMSE as performance criteria. Table 4 shows that the correlation is neither strong nor consistent across settings. Thus the alignment between $d_{\text{MPNN}}$ and $d_{\text{struc}}$ is not a key to high MPNN performance. This answers **Q1.5** negatively.

Table 4: The correlation coefficient between $\text{RMSE}(d_{\text{MPNN}}, d_{\text{struc}})$ and the performance on the training and test sets. Performance was measured based on accuracy for Mutagenicity and ENZYMES, and based on RMSE for Lipophilicity.

| | | Train | | | | Test | | | |
|---|---|---|---|---|---|---|---|---|---|
| | | GED | TMD | WLOA | WWL | GED | TMD | WLOA | WWL |
| Mutagenicity | mean | 0.26 | 0.22 | -0.06 | 0.43 | 0.31 | 0.31 | 0.06 | 0.50 |
| | sum | 0.04 | 0.23 | 0.35 | 0.30 | -0.09 | 0.17 | 0.20 | 0.29 |
| ENZYMES | mean | 0.32 | 0.28 | 0.24 | 0.29 | 0.37 | 0.53 | 0.38 | 0.20 |
| | sum | -0.35 | 0.68 | 0.41 | 0.32 | -0.53 | 0.13 | -0.09 | -0.11 |
| Lipophilicity | mean | -0.67 | -0.65 | -0.66 | -0.56 | -0.59 | -0.67 | -0.59 | -0.59 |
| | sum | -0.11 | -0.60 | -0.52 | -0.30 | -0.40 | -0.82 | -0.77 | -0.58 |
| IMDB-BINARY | mean | 0.04 | 0.18 | -0.38 | -0.31 | -0.26 | -0.24 | -0.35 | 0.37 |
| | sum | 0.41 | 0.67 | -0.62 | -0.60 | 0.07 | 0.21 | -0.19 | -0.07 |
| COLLAB | mean | 0.75 | 0.63 | 0.59 | -0.54 | 0.67 | 0.54 | 0.53 | -0.43 |
| | sum | -0.47 | 0.56 | -0.50 | -0.55 | -0.36 | 0.48 | -0.38 | -0.48 |

Table 5: The mean±std of $\text{RMSE}(d_{\text{MPNN}}, d)$ [$\times 10^{-2}$] over five different seeds. Each column corresponds to GIN with a given graph pooling method, trained on a given dataset.

| | Mutagenicity | | ENZYMES | | Lipophilicity | |
|---|---|---|---|---|---|---|
| | mean | sum | mean | sum | mean | sum |
| $d_{\text{WWL}}$ | 11.47±0.24 | 14.08±0.77 | 11.54±0.30 | 12.10±0.84 | 14.12±0.60 | 14.97±0.58 |
| $d_{\text{WLOA}}$ | 17.99±2.79 | 13.05±1.44 | 23.71±0.81 | 9.94±1.88 | 16.95±0.52 | 13.97±0.75 |
| $\dot{d}_{\text{WILT}}$ | **3.70 ± 0.57** | 3.86±0.40 | **5.32 ± 0.20** | 8.60±0.35 | **6.31 ± 0.46** | **6.49 ± 0.50** |
| $\bar{d}_{\text{WILT}}$ | 4.98±0.78 | **3.56 ± 0.36** | 7.55±0.24 | **3.86 ± 0.68** | 9.52±0.70 | 6.59±0.51 |

# F    EXPERIMENTAL DETAILS FOR SECTION 5

For the experiments in Section 5, we trained 3-layer GCN and GIN with embedding dimensions of 64 on the three datasets. We explored both mean and sum pooling. Each model was trained on the full dataset for 100 epochs using the Adam optimizer with a learning rate of $10^{-3}$. Then, each model was distilled to WILT by minimizing the loss $\mathcal{L}$ defined in Section 4.4. We used the entire data set for $\mathcal{D}$ in $\mathcal{L}$. The distillation was done using gradient descent optimization with the Adam optimizer for 10 epochs. The learning rate and batch size were set to $10^{-2}$ and 256, respectively. See Algorithm 1 for details.

In Table 2, we only show the results for GCN. Here, we show results for GIN in Table 5. The overall trend is the same between Tables 2 and 5: $\dot{d}_{\text{WILT}}$ and $\bar{d}_{\text{WILT}}$ are much better aligned with $d_{\text{MPNN}}$ than $d_{\text{WWL}}$ and $d_{\text{WLOA}}$. In addition, $d_{\text{WWL}}$ and $\dot{d}_{\text{WILT}}$ approximate $d_{\text{MPNN}}(\text{mean})$ better, while the opposite is true for $d_{\text{MPNN}}(\text{sum})$. We also observed the same trend on the IMDB-BINARY dataset (see Table 6).

Next, we plot the distribution of WILT edge weights after distillation in Figure 8. While the range of edge weights varies by model and dataset, all the distributions are skewed to zero (note that the y-axis is log scale). This suggests that only a small fraction of all WL colors influence $d_{\text{MPNN}}$. In other words, MPNNs build up their embedding space based on a small subset of entire WL colors, regardless of model and dataset.

Finally, we visualize the WL colors with the largest weights, i.e., whose presence or absence influence $d_{\text{WILT}}$ and therefore – by approximation – $d_{\text{MPNN}}$ the most. We use the Mutagenicity dataset as functionally important substructures are known from domain knowledge (Kazius et al., 2005). It should be noted that we only consider colors that appear in at least 1% of all graphs in the dataset. Table 7 and 8 show graphs with substructures corresponding to the WL colors with the top 10 largest weights. Table 7 is the result for GCN with sum pooling, while Table 8 is for GCN with mean pooling. If the highlighted subgraph matches one of the seven toxicophore substructures listed in Table 1 of Kazius et al. (2005), we show the toxicophore name as well. 4 and 3 out of 10 WL colors cor-

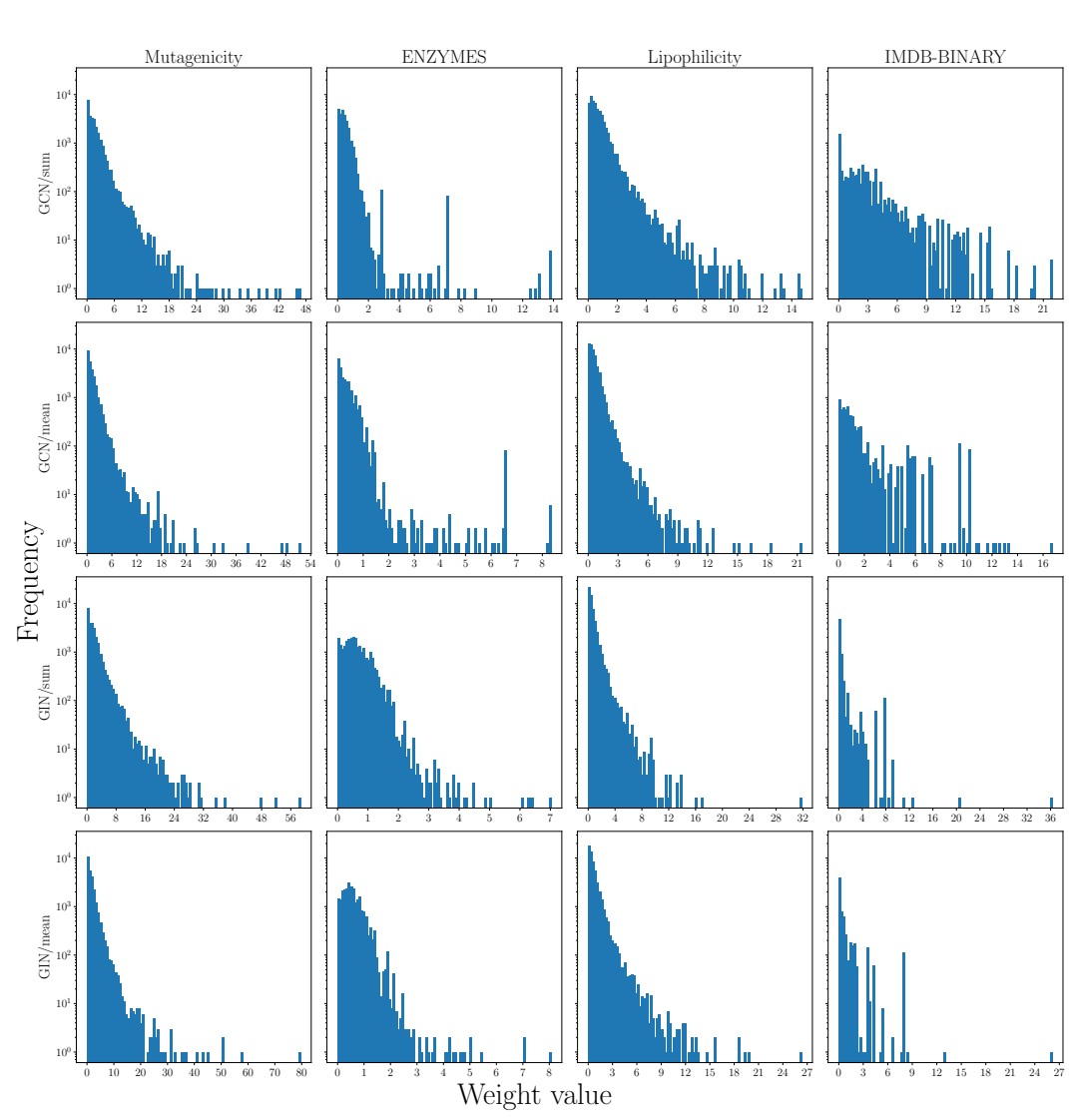

Figure 8: The distribution of edge weights of WILT after distillation from varying models trained on different datasets. The models with sum pooling were distilled into WILT with dummy normalization, while the models with mean pooling were distilled into WILT with size normalization. The log scale y-axis is shared across all plots.

Table 6: The mean±std of RMSE($d_{MPNN}, d$) [$\times 10^{-2}$] over five different seeds. Each column corresponds to a GCN or GIN with a given graph pooling method, trained on IMDB-BINARY.

| | GCN | | GIN | |
|---|---|---|---|---|
| | mean | sum | mean | sum |
| $d_{WWL}$ | 16.98±2.06 | 16.21±2.45 | 21.32±0.25 | 23.49±0.42 |
| $d_{WLOA}$ | 19.04±4.39 | 12.01±3.81 | 19.65±0.45 | 21.23±0.39 |
| $\dot{d}_{WILT}$ | **6.19 ± 1.24** | 9.08±4.37 | **2.61 ± 0.34** | 8.09±0.89 |
| $\bar{d}_{WILT}$ | 7.62±1.27 | **4.69 ± 3.70** | 3.09±0.37 | **0.85 ± 0.13** |

Table 7: Example graphs with highlighted significant subgraphs corresponding to colors with top 10 largest weights. GCN with sum pooling was used. The toxicophore name is shown if the highlighted subgraph matches toxicophore substructures reported in Table 1 of Kazius et al. (2005)

| (1) three-membered heterocycle (epoxide) | (2) | (3) | (4) alphatic halide | (5) |
|---|---|---|---|---|

| (6) nitroso | (7) | (8) | (9) | (10) alphatic halide |
|---|---|---|---|---|

respond to toxicophore substructures in Tables 5 and 6, respectively, which is quite a lot considering that only 7 toxicophore substructures are listed in Table 1 of Kazius et al. (2005). Furthermore, there are some colors that not fully but partially match one of the substructures in Kazius et al. (2005). For instance, (6) and (9) in Table 7 and (8) in Table 8 partially match "aromatic nitro", while (7) in Table 8 is part of "polycyclic aromatic system". Note that it is impossible to identify subgraphs that perfectly match these toxicophore substructures, since our method can only identify subgraphs corresponding to a region reachable within fixed steps from a root node. For example, the subgraph in (1) of Table 7 is a region reachable in 2 steps from the oxygen O. This limiation may seem to be a drawback of our proposed method, but in fact it is not. It is natural to identify only subgraphs corresponding WL colors to interpret $d_{MPNN}$, because MPNNs can only see input graphs as a multiset of WL colors.

1350
1351
1352
1353
1354
1355
1356
1357
1358
1359
1360
1361
1362
1363
1364
1365
1366
1367
1368
1369
1370
1371
1372
1373
1374
1375
1376
1377
1378
1379
1380
1381
1382
1383
1384
1385
1386
1387
1388
1389
1390
1391
1392
1393
1394
1395
1396
1397
1398
1399
1400
1401
1402
1403

Table 8: Example graphs with highlighted significant subgraphs corresponding to colors with top 10 largest weights. GCN with mean pooling was used. The toxicophore name is shown if the highlighted subgraph matches toxicophore substructures reported in Table 1 of Kazius et al. (2005)

| (1) | (2) three-membered heterocycle (epoxide) | (3) alphatic halide | (4) | (5) |
|---|---|---|---|---|

$CH_2Cl$
$CH_2Cl-C-CH_2Cl$
$CH_2Cl$

H   O   H

$CH_3-CH$, Br, Br

P (triphenylphosphine)

N, $NO_2$

| (6) | (7) | (8) nitroso | (9) | (10) |
|---|---|---|---|---|

$O-CH_3$
$CH_3-C-CH_2-CH_3$
$CH_3$

N

S, N, $CH_2$, OH, O

$NO_2$, C, N

O

