# OpenReview forum: "WILTing Trees: Interpreting the Distance Between MPNN Embeddings"
_ICLR.cc/2025/Conference — Submitted to ICLR 2025_

### Official Review · Reviewer_DKn5 · 2024-10-16

**Soundness:** 1
**Presentation:** 1
**Contribution:** 1
**Rating:** 1
**Confidence:** 5

**Summary:**

The paper analyzes several graph metrics in order to evaluate model performance and metric preservation.

**Strengths:**

The paper analyzes several graph metrics and sees correspondence between metrics on graphs and metrics on datasets and on MPNNs.

**Weaknesses:**

The paper lacks novelty, as it presents neither a new analysis nor the introduction of a new network. Its contributions fall short of the expectations for an ICLR-style conference, where higher levels of innovation and original research are typically required.
Specifically, Definition 4 (Evaluation Criterion for Alignment Between dMPNN and dfunc) doesn't capture any alignment between MPNN and func. Usually, the ratio of MPNN(G) -MPNN(H) and struct(G,H) is measured, and in this case, some previous papers showed theoretically, this ratio converges to zero for a specific sequence of graphs. High/Low of your proposed measure doesn't intuately mean something.
Genreally I really don't see any novelty or something surprising in this paper.

**Questions:**

Why did you take Definition 4 (Evaluation Criterion for Alignment Between dMPNN and dfunc) as a measure? What does it mean?

---

> ### Author Response · Authors · 2024-11-15
>
> Thank you for taking the time to review and comment on our paper. We clarify our contribution and answer your question below.
>
> ---
>
> > The paper lacks novelty, as it presents neither a new analysis nor the introduction of a new network.
> >
>
> We respectfully disagree. The proposed WILT generalizes both WLOA distances by Kriege et al (2016) and WWL distances by Togninalli et al (2019). It introduces weights to these distances and presents a novel way to train these weights. Up to our knowledge, WLOA and WWL distances have so far only been presented and used without tunable weights, i.e., as structural distances. Our proposal now allows to fit them to target data.
>
> In terms of analysis, we experimentally investigate what the $d_{MPNN}$ of trained MPNNs looks like. We clarifiy that MPNNs are trained in a way that $d_{MPNN}$ respects the task-relevant functional distance $d_{func}$ rather than the task-irrelevant structural distance $d_{struc}$. In addition, by fitting the WILT distance $d_{WILT}$ to $d_{MPNN}$, we identified Weisfeiler Leman subgraphs (colors) that determine $d_{MPNN}$, providing new insights into $d_{MPNN}$.
>
> Please read the official comment to all reviewers, where we clarify our contributions.
>
> > Specifically, Definition 4 (Evaluation Criterion for Alignment Between dMPNN and dfunc) doesn't capture any alignment between MPNN and func.
> >
>
> In Definition 4, $A_k(G)$ and $B_k(G)$ measure the average functional distance between $G$ and its neighbors/non-neighbors in the embedding space, respectively. Therefore, $A_k(G) < B_k(G)$ means that graphs functionally similar to $G$ tend to be embedded closer to $G$ than functionally dissimilar graphs. Therefore, the larger $ALI_k$ is, the more we say that $d_{MPNN}$ is aligned with $d_{func}$. This measure is related to the performance of a k-Nearest-Neighbor model on $d_{MPNN}$.
>
> While the ratio of two distances is commonly used to measure the correspondence between them, there are two reasons why we don't use it in Definition 4.
>
> 1. When the task is graph classification, $d_{func}$ is a binary function that returns 1 if two graphs belong to the same class, otherwise 0. Thus, it is unreasonable to expect the exact match of real-valued $d_{MPNN}$ and binary $d_{func}$.
> 2. Even when the task is graph regression, it is natural to expect that the scale(min/max) of $d_{MPNN}$ and that of $d_{func}$ are different. We can think of normalizing them and measuring the ratio, but for consistency with the classification case, we don't use such a metric.
>
> ---
>
> Thank you again for your review and comments. We hope our explanation addresses your concerns. Please let us know if you have any further questions.

---

### Official Review · Reviewer_LXNE · 2024-10-18

**Soundness:** 3
**Presentation:** 4
**Contribution:** 2
**Rating:** 6
**Confidence:** 3

**Summary:**

This paper suggests a way of understanding how MPNNs model the graph functional distance. Specifically, the author distills MPNNs into their proposed Weisfeiler Leman Labeling Tree (WILT) without changing the graph distance. The proposed algorithm operates on linear time, which yields optimal transport distance between Weisfeiler Leman histograms. Empirical analysis shows that the relative position of the embedding follows the importance of specific subgraphs, showing that MPNNs can capture domain knowledge.

**Strengths:**

**S1.** This work provides theoretical contribution in understanding the relationships between the MPNNs and structural distance.

**S2.** Some illustrative examples, e.g., from Figure 1 to 3 improve the readability of the manuscript.

**S3.** Extensive experiments show the validity of the proposed insight.

**Weaknesses:**

**W1.** This paper aims to show how Message Passing Neural Networks (MPNNs) define the relative position of embeddings. Starting from Definition 5, this manuscript suggests the WILTing Distance, which modifies the distance metric in Optimal Transport (OT) as d_path and stems from the mechanism of the shortest path metric on a tree [2]. Additionally, the author employs [3] for efficient (linear) computation. However, most of their contributions overlap with prior work [1], which proved that MPNNs have the same expressive power as the 1-Weisfeiler-Lehman (1-WL). From my viewpoint, the contribution of this work seems to be marginal unless it is compared with [1] properly.
  * [1] Fine-grained Expressivity of Graph Neural Networks, NeurIPS '23
  * [2] Fast subtree kernels on graphs, NeurIPS '09
  * [3] Wasserstein weisfeiler-lehman graph kernels, NeurIPS '19

Q1) Could you please elaborate on the difference between [1] and your work?


**W2.** Most of my concern lies on the above question since the writing of this paper is very clear and the experiments are also interesting.

Q2) Could you please add [1] to the experiments as well?


I'm willing to increase the score if the above concern is addressed clearly.

**Questions:**

Please see the weaknesses above.

---

> ### Author Response · Authors · 2024-11-15
>
> We thank you for your thorough review and valuable questions about the connection between [1] and our work. Below are the answers to your two questions.
>
> ---
>
> > Q1: most of their contributions overlap with prior work [1], which proved that MPNNs have the same expressive power as the 1-Weisfeiler-Lehman (1-WL).
> >
>
> While the distance functions introduced in [1] are related to our work, there are important differences that make their framework difficult to apply in our context. [1] proposes task-irrelevant structural distance measures that are small for graphs $G, H$ if and only if all MPNNs compute representations for $G,H$ that are close. In contrast, our analysis focuses on how $d_{MPNN}$ of a single MPNN captures the task-relevant functional distance of grpahs. Moreover, [1] deals only with dense graphs, while our method works with practical sparse graphs.
>
> > Q2: Could you please add [1] to the experiments as well?
> >
>
> In addition to the above difference, the time complexity of graph distances proposed in [1] seems to be O($n^5 \log n$)~O($n^7$), which is too demanding in practice. Nevertheless, we are currently looking for a way to include their distances in our experiments, since there is a connection between [1] and our study.
>
> ---
>
> Thank you again for your valuable feedback. Please also read to our official comment to all reviewers, where we clarify our contributions. Please let us know if you have any more suggestions or questions.

---

> > ### Comment · Reviewer_LXNE · 2024-11-26
> > **Thank you for the rebuttal**
> >
> > Dear authors,
> >
> > Thank you for addressing my concerns; I have updated my score accordingly.

---

### Official Review · Reviewer_ys1y · 2024-10-24

**Soundness:** 2
**Presentation:** 1
**Contribution:** 2
**Rating:** 3
**Confidence:** 3

**Summary:**

This paper investigates the distance of MPNN embeddings. The authors empirically found that the Euclidean distance of MPNN embeddings after training is aligned with the Euclidean distance of the graph labels. The authors then proposed a new graph pseudometric --- WILTing Distance --- for distilling MPNN embedding distance. The authors showed experimentally that the proposed WILTing Distance approximates the MPNN distance well, while revealing the important subgraph structure for the molecule property prediction tasks.

**Strengths:**

1. The study of graph distance and connection with graph neural networks is of high interest to the community.

2. The figures are nicely rendered.

**Weaknesses:**

1. The organization of the paper is hard to follow. The paper seems to have two independent parts. The first half (Sec 3 and 4) investigates the MPNN distance by comparing it with graph structural distances (task-independent) versus graph label distances (task-dependent). The second half (Sec 5 and 6) aims to distill the MPNN distance into the proposed WILTing distance.

2. Unclear motivation. The first half seems rather intuitive: the MPNN embeddings (and thus their distances) are optimized to predict the target graph labels (in both classification and regression) and thus align with the target distances; the authors should justify and discuss more thoroughly why Q2-Q5 worth investigation. The second half touches on a few interesting aspects (e.g., optimal transport, distance upper bounds, MPNN interpretability, etc), but the authors did not connect them in a coherent way, nor dive deep in any of them.

3. Limited contribution: The property of the proposed WILTing distance, and its connections with other recently proposed distances are not thoroughly discussed. See more details in the Questions.

**Questions:**

1. The purpose of the WILTing distance is to identity the important (learned) WL colors that strongly influences the MPNN distance. This can in turn be used to identify important edges or subgraphs that matters for the downstream task, providing a tool for MPNN interpretability. Is MPNN interpretability the main practical motivation of WILTing distance?
(a) If so, why not compare the important subgraphs identified from WILTing distance with other GNN interpretability tools (e.g. [1],[2]). What are the additional insights or advantages from using WILTing distance over existing interpretability tools?
(b) If not, what are other motivations of WILTing distance? Can it be a drop-in replacement of MPNN?

2. Expressivity of WILTing distance (Appendix B.4): The authors define $d_{\text{WL}}$ using the binary notion of expressivity in terms of distinguishing non-isomorphic graphs. However, recent works in [3], [4] have proposed a fine-grained, continuous notion of WL distances based on optimal transport of the induced measures of the WL colors, and the relationship between the continuous WL distances with the MPNN distance. It seems more natural and stronger to investigate the expressivity of WILTing distance under the continuous WL distance. Can the authors justify their definition and comment on the expressivity of WILTing distance compared to the continuous WL distance?

3. Relationship between WILTing distance and Tree Mover Distance [5]: The authors discuss the connections between WILTing distance and the graph edit distance (Thm 1) as well as Weisfeiler Leman Optimal Assignment distance (Thm 2). Intuitively, WILTing distance seems very similar to the Tree Mover Distance [3]: Can the authors compare them?




References:

[1] Yuan, Hao, et al. "On explainability of graph neural networks via subgraph explorations." International conference on machine learning. PMLR, 2021.

[2] Ying, Zhitao, et al. "Gnnexplainer: Generating explanations for graph neural networks." Advances in neural information processing systems 32 (2019).

[3] Chen, Samantha, et al. "Weisfeiler-lehman meets gromov-Wasserstein." International Conference on Machine Learning. PMLR, 2022.

[4] Böker, Jan, et al. "Fine-grained expressivity of graph neural networks." Advances in Neural Information Processing Systems 36 (2024).

[5] Ching-Yao Chuang and Stefanie Jegelka. Tree mover’s distance: Bridging graph metrics and stability of graph neural networks. Advances in Neural Information Processing Systems, 2022.

---

> ### Author Response · Authors · 2024-11-15
>
> We thank you for your thorough review and valuable suggestions. We address each weakness and question below.
>
> ---
>
> > W1~W3: Hard to follow organization, unclear motivation, limited contributions.
> >
>
> We have explained the motivation, logical flow, and our contributions in the official comment to all reviewers. We will post the updated paper next week, where we will clarify the importance of answering Q1-Q5, and discuss the theory and algorithm of WILT in detail in the main text, which is currently in Appendix B.
>
> > Q1:  Is MPNN interpretability the main practical motivation of WILTing distance? (a) If so, why not compare the important subgraphs identified from WILTing distance with other GNN interpretability tools (e.g. [1],[2]). What are the additional insights or advantages from using WILTing distance over existing interpretability tools?
> >
>
> Yes, we introduce the WILTing distance for interpreting MPNNs. However, the motivation of our work and the previous studies such as [1, 2] are quite different. Our goal is to understand the entire metric structure $d_{MPNN}$ by identifying Weisfeiler Leman subgraphs that determine $d_{MPNN}$. On the other hand, [1, 2] aim to find an instance-level explanation for the prediction of one input graph. This difference in global-level/distance vs instance-level/prediction makes it difficult to compare our method with [1, 2]. To the best of our knowledge, this is the first work to analyze entire $d_{MPNN}$ in terms of WL subgraphs, and provides new insights such as:
>
> - $d_{MPNN}$ is determined by only a small fraction(~5%) of the entire set of Weisfeiler Leman (WL) colors.
> - The identified WL colors are also known to be important by domain experts.
>
> > Q2: Expressivity of WILTing distance (Appendix B.4): The authors define using the binary notion of expressivity in terms of distinguishing non-isomorphic graphs. However, recent works in [3], [4] have proposed a fine-grained, continuous notion of WL distances […] Can the authors justify their definition and comment on the expressivity of WILTing distance compared to the continuous WL distance?
> >
>
> While the distance functions introduced in [3,4] are related to our work, there are important differences that make these frameworks difficult to apply in our context. [4] proposes structural distances $d, d’$ that are small for graphs $G, H$ if and only if *all MPNNs compute representations for $G,H$ that are close.* Our analysis focuses on a single MPNN and its resulting distance $d_{MPNN}$, which is not covered by their analysis. In particular, if $d_{MPNN}$ is small, $d, d’$ may be large. The WL-distance proposed in [3], again is a structural metric and can not be adapted to analyze a given MPNN.
>
> Regarding relaxation of expressivity, [3, Prop 3.3] shows similar results  than we do: Their $d_{WL}(G,H) = 0 \Leftrightarrow G,H$ are WL-distinguishable. In this (binary) sense, $d_{WILT}$ is as expressive as $d_{WL}$, if the same initial node labels are used.  [3] proposes to use their distance function as a relaxation of expressivity. Alternatively, our $d_{WILT}$ can be used in the same way. However, a quantitative analysis of the similarities between $d_{WL}$ and $d_{WILT}$ is beyond the scope of this work.
>
> > Q3: Relationship between WILTing distance and Tree Mover Distance [5]: […]. Intuitively, WILTing distance seems very similar to the Tree Mover Distance [5]: Can the authors compare them?
> >
>
> Both the WILTing distance $d_{WILT}$ and the Tree Mover's Distance $d_{TMD}$ are optimal transport distances between multisets of Wesifeiler Leman (WL) subgraphs. The difference lies in how they define the ground metric (cost) between each pair of WL subgraphs: $d_{WILT}$ adopts the shortest path length on WILT, while the $d_{TMD}$ uses recursive optimal transport of WL subgraphs (Definition 4 in [5]). As a result:
>
> - $d_{WILT}$ can be computed in O($|V|$), while $d_{TMD}$ requires O($|V|^3 \log|V|$).
> - $d_{WILT}$ has tunable edge parameters, while $d_{TMD}$ does not. So, $d_{WILT}$ is suitable for approximating $d_{MPNN}$.
> - In terms of binary expressive power, $d_{WILT}$ (dummy normalization) = $d_{TMD}$ = 1-WL test
>
> One advantage of $d_{TMD}$ is that it can handle continuous node festures, while $d_{WILT}$ cannot. But understanding $d_{MPNN}$ of graphs with continuous node features is beyond the scope of this study.
>
> ---
>
> Thank you again for your valuable feedback, which helps us improve the clarity and contribution of our work. We hope our explanation clearly addresses your concerns and questions. Please let us know if you have any more questions or suggestions.

---

> > ### Comment · Reviewer_ys1y · 2024-11-24
> >
> > I thank the authors for their explanations.
> >
> > Follow-ups:
> > - Updated manuscript:
> >   - I don't see it on OpenReview yet. I strongly encourage the authors to post it as soon as possible
> > - Q1: “This difference in global-level/distance vs instance-level/prediction makes it difficult to compare our method with [1, 2]”
> >   - While I agree [1,2] is an instance-level interpretability result, I don’t follow why WILTing distance provides global-level result. Specifically, the identified small fraction of WL colors are input-graph dependent, as shown in Fig.5

---

> > > ### Author Response · Authors · 2024-11-25
> > >
> > > Thank you for your further question, and sorry for the delay. We have posted a new version of the pdf, and summarized the main updates in the comment to all reviewers.
> > >
> > > > Q1: I don’t follow why WILTing distance provides global-level result. Specifically, the identified small fraction of WL colors are input-graph dependent, as shown in Fig.5
> > >
> > > $d_{WILT}$ offers an interpretation by finding WL colors $c$ s.t. “If $\textbf{any}$ $G$ and $H$ have different numbers of nodes corresponding to $c$, $d_{MPNN}(G, H)$ is large".
> > > So, our interpretation is not about specific graphs, but about all graphs.
> > > This comes from the fact that the edge weights $w$ of WILT are shared among all graphs in a dataset.
> > > In Figure 5 of the old version (Figure 4 of the current version), we show graphs having $c$ with a large $w_c$. But this is just for visualizing the identified $c$. The weight $w_c$ and the identified $c$ are instance-independent.
> > >
> > > We hope our response addresses your concerns. If it is still unclear, or if you have other questions, we are happy to answer them.

---

> > > > ### Comment · Reviewer_ys1y · 2024-12-02
> > > > **Follow-up on the different metrics for alignment (ALI versus RMSE)**
> > > >
> > > > I thank the authors for their revision of the paper. A quick follow-up question:
> > > > In Line 154-157 and Appendix E (analysis for d_struc): The authors claim that d_func is crucial for MPNN's performance whereas d_struc is not. However, they discuss that their chosen metrics for measuring the alignment between d_mpnn and d_func (ALI in Defn 3) is different from d_struc (RMSE, Dean 12). As such, although the correlation is higher for d_func, it does not fully prove that d_func is crucial for MPNN’s performance whereas d_struc is not (e.g., the smaller correlation could be due to the choice of RMSE compared to ALI). Do I miss anything? If not, I recommend the authors to weaken their claims.

---

> > > > > ### Author Response · Authors · 2024-12-03
> > > > >
> > > > > Thank you for your additional question. As you point out, we use two different metrics to measure the alignment between $d_{MPNN}$ and $d_{struc}$ or $d_{func}$ due to the binary functional and the nonbinary structural pseudometrics. Therefore, it is inappropriate to compare the values of the two measurements directly, and we apologize for any misleading statements that may sound like we are directly comparing them.
> > > > >
> > > > > However, we still argue that the alignment between $d_{MPNN}$ and $d_{struc}$ is less relevant to MPNN performance than the alignment between $d_{MPNN}$ and $d_{func}$ by investigating the consistency of the correlation over multiple datasets. In Tables 1 and 3, the correlation between $ALI_k (d_{MPNN}, d_{func})$ and accuracy is always positive for classification and always negative for regression, with only one exception in IMDB-BINARY. In Table 4, however, there are inconsistent results (positive correlation between $\text{RMSE}(d_{\text{MPNN}}, d_{\text{func}})$ and accuracy for classification and negative correlation for regression) in all datasets. In addition, the ALI is consistently improved by training (Figure 2, 5), while the RMSE is not (Figure 7). Thus, we conclude that the alignment to $d_{func}$ is more important for MPNN performance than the alignment to $d_{struc}$.
> > > > >
> > > > > Thank you again for your engagement during the rebuttal period.

---

### Official Review · Reviewer_6hU2 · 2024-11-04

**Soundness:** 3
**Presentation:** 2
**Contribution:** 4
**Rating:** 5
**Confidence:** 2

**Summary:**

This paper explores the metric properties of the embedding space in message passing neural networks. The authors observe that the embedding distances of MPNNs align with the functional distances between graphs, contributing to the predictive power of these networks. The primary contribution is the proposal of a Weighted Weisfeiler Leman Labeling Tree (WILT), which distills MPNNs while preserving graph distances. This WILT framework enables interpretable optimal transport distances between Weisfeiler Leman histograms, improving the interpretability of MPNNs by identifying subgraphs that influence embedding distances.

**Strengths:**

- As far as I am aware, the introduction of WILT to interpret MPNN embedding spaces is unique. By distilling MPNNs into WILT, the method is able to understand the role of specific subgraphs in determining functional distances. This seems especially useful in settings where interpretability is important.
- By showing that MPNN embeddings naturally align with functional graph distances, WILT provides insight into why MPNNs achieve high predictive accuracy in certain tasks. This contribution enhances the field’s understanding of how MPNNs implicitly capture task-relevant structures in the embedding space, perhaps even opening the way for 'transferring' this knowledge. Moreover,  by offering a framework that generalizes high-performance kernels, the paper opens doors for developing kernels tailored to specific graph applications.
- WILT generalizes existing Weisfeiler Leman approaches. As these approaches are used in a wide variety of tasks, e.g. molecular prediction, making WILT a versatile tool. I especially like the approach runs in linear time, making it also applicable for e.g. large molecules.
- I really appreciate the figures in the paper.

**Weaknesses:**

- Even though I appreciate the theoretical contributions of this paper, I think it would benefit significantly from more high-level intuition of the approach. The introduction is very short, and the paper is very condensed, providing little guidance for the reader. I would really urge the authors to move part of the formalism to the appendix and dedicate more space in the paper to building intuition behind the approach, as this is to me the major weakness in the paper.
- The paper would benefit from a more thorough comparison to recent interpretability approaches in graph learning, such as methods that use attention mechanisms or explainable subgraph extraction. I think this could really highlight the differences and benefits of this approach.
- The empirical validation is limited and its effectiveness on other types of graphs (e.g., social networks, knowledge graphs) is not thoroughly explored. In molecular prediction tasks, we know that the topological information of the graph is very indicative of the predicted properties, but how beneficial is this work in these more subtle settings? Some non-molecular exploration would be hugely beneficial to judge the applicability of the framework.
- There is a lot of work on extending the WL test to higher-order (e.g. simplicial, cellular etc). As WILT inherits the typical limitations of the WL test, it could perhaps benefit from these higher-order topological spaces, as the authors mention. This is claimed to be straight-forward, but some deeper reflections on this would be beneficial.

**Questions:**

- Have the authors considered adapting WILT to higher-order Weisfeiler Leman test? Or maybe using alternative graph matching approaches?
- Given the efficiency of WILT, did the authors consider testing its scalability on high-dimensional datasets, e.g. social networks or molecular interaction networks? This would help demonstrate the method’s robustness across diverse graph types.
- Could the authors expand on how WILT compares to other interpretability methods in terms of capturing functional subgraphs, e.g. those based on attention?
- Can WILT work with incomplete graphs at all? What about directed graphs?

---

> ### Author Response · Authors · 2024-11-15
>
> We thank you for your thorough review and valuable suggestions. We hope the points below adequately address your questions.
>
> ---
>
> > W1: Lack of high-level intuition, guidance.
> >
>
> We have explained the motivation, logical flow, and our contributions in the official comment to all reviewers. We will post the updated paper next week, where we will clarify the motivation and logical flow, and add more intuitive explanations.
>
> > W2, Q3: Comparison to recent interpretability approaches in graph learning, such as methods that use attention mechanisms or explainable subgraph extraction
> >
>
> Although we introduce the WILTing distance for interpreting MPNNs, the motivation of our work and the previous interpretation methods are quite different. Our goal is to understand the entire metric structure $d_{MPNN}$ by identifying Weisfeiler Leman subgraphs that determine $d_{MPNN}$. On the other hand, previous studies aim to find an instance-level explanation for the prediction of one input graph. This is true for both subgraph extraction methods and attention analysis methods. This difference in global-level/distance vs instance-level/prediction makes it difficult to compare our method with previous interpretation methods. To the best of our knowledge, this is the first work to analyze entire $d_{MPNN}$ in terms of WL subgraphs, and provides new insights such as:
>
> - $d_{MPNN}$ is determined by only a small fraction(~5%) of the entire set of Weisfeiler Leman (WL) colors.
> - The identified WL colors are also known to be important by domain experts.
>
> > W3, Q3: The empirical validation on other types of graphs (e.g., social networks, knowledge graphs, molecular interaction networks)
> >
>
> We are currently running experiments on the non-molecular IMDB dataset and will report the results next week. It should be noted that our WILTing distance is designed to analyze the distance between graph embeddings. Therefore, very large networks such as social networks, where the main focus is on node prediction/embedding are beyond the scope of this work.
>
> > W4, Q1: Have the authors considered adapting WILT to higher-order Weisfeiler Leman test? Some deeper reflections on this would be beneficial.
> >
>
> Theoretically, it is straightforward to extend WILT to a higher-order WL test. We can build a WILT from the results of the higher-order WL test in exactly the same way as in the 1-WL case, whenever the higher-order test results in a hierarchy of labels (which all ‘generalized WL-tests’ that we are aware of do). That is: generalized WL-tests update labels (not necessarily of nodes, but of higher order objects such as cell complexes, or sets of nodes, …) iteratively. Each label in iteration $t$ has a unique parent label in iteration $t-1$. The WILT on such a generalized hierarchy has the same expressivity as the hierarchy itself. Our proofs in the appendix hold in these cases, as well.
>
> In future studies, it would be interesting to analyze higher-order variants of MPNNs by WILT with corresponding expressiveness. However, there may be a practical difficulty, since the number of trainable edge weights is expected to be increase with increasing order.
>
> > Q4: Can WILT work with incomplete graphs at all? What about directed graphs?
> >
>
> What do you mean by "incomplete graphs"? If you mean a graph with non-adjacent node pairs, the answer is yes. WILT makes no assumptions about the topological structure of graphs. If you mean missing node or edge labels, one way would be to impute the missing labels using some separate technique.
>
> The extension to directed graphs is easy. All you need to do is run the corresponding variant of the WL test when building WILT, which updates the color of node $v$ only based on the colors of $u$ with an edge $u \to v$.
>
> ---
>
> Thank you again for your feedback, which helps us clarify our contributions. In addition, your questions let us consider extending our work to higher-order MPNNs and broader types of graphs. Please let us know if you have any further questions or suggestions, and we will be happy to answer them.

---

> > ### Comment · Reviewer_6hU2 · 2024-11-24
> >
> > Dear authors. Thanks a lot for the comments. Since my primary concerns are regarding the presentation, readability, and structure of the paper, I will maintain my score as thus far no new manuscript has been uploaded.

---

> > > ### Author Response · Authors · 2024-11-24
> > >
> > > We thank you for reading our answers, and apologize for the delay. We have now uploaded a new pdf. We have explained the main updates in the comment to all reviewers above. If there are still parts that are unclear, please let us know. Further questions and comments are also welcome.

---

> > > > ### Comment · Reviewer_6hU2 · 2024-12-03
> > > >
> > > > Thanks a lot for the updated manuscript.
> > > >
> > > > Even though the readability has somewhat improved, I think the paper could use one more round of fine-tuning before publication. Also reading the other reviewers' comments, I will maintain my score.

---

> > > > > ### Author Response · Authors · 2024-12-03
> > > > >
> > > > > Thank you again for your comments. We have addressed all your questions and suggestions. Especially, we improved the presentation, which was your main concern. If it is not too much to ask, we would like to get some pointers (after the reviewer-AC discussion phase) towards the specific parts of the paper that need more fine-tuning to improve our paper in the future even if it does not get accepted to ICLR.

---

### Official Review · Reviewer_H8oL · 2024-11-04

**Soundness:** 4
**Presentation:** 3
**Contribution:** 4
**Rating:** 8
**Confidence:** 4

**Summary:**

The paper aims to shed light on the workings of GNNs by investigating in how far the distance given by the graph embedding of the GNN is reflected in other graph distances. The authors find that the MPNN distance is not correlated to static graph distances that are oblivious to the task. However, it is related to the "functional" distance, which encodes the class label. The authors propose a novel technique, the WILT. The WILT is a tree, whose nodes are the colors of WL and whose edges connect preceeding colors to their successors in the iterations of WL. The WILT can be tailored to a specific problem by learning weights on the edges. The authors find high correlation between the WILT and MPNN performance.

**Strengths:**

- The paper provides a solid theoretical basis for the proposed methods, including proofs and detailed explanations of pseudometrics.
- By identifying important subgraphs, the paper enhances the interpretability of MPNNs, making it easier to understand what drives their performance.

**Weaknesses:**

- The experiments are conducted on specific datasets; it would be beneficial to see more diverse real-world applications to assess generalizability.
- The answers to the questions asked are pretty obvious beforehand. The structural distances that stem from non-trainable graph kernels have nothing to do with the task, therefore it is unreasonable to assume that an MPNN (before or after training) would be highly correlated (Q2, Q3) . The same goes for Q4, Q5, where the functional distance encodes the target, and is therefore what the MPNN is optimized for. While it is not inherently bad to ask questions that one expects the answer to, these questions, though many, create little new insight.
- The algorithm for learning the WILT weight is only discussed in the appendix.

**Questions:**

- How expressive is WILT? It implies a hyperbolic distance between colors, so intuitively, it should be weaker than MPNNs?
- How long does learning the WILT weights take?
- Famously WL is extremely sensitive to noise in the graph structure. Does WILT handle structural noise and/or feature noise well?

---

> ### Author Response · Authors · 2024-11-15
>
> We thank the reviewer for their thorough review, acknowledging our contributions, and voting to accept our paper. We address each weakness and question individually below.
>
> ---
>
> > W1: The experiments are conducted on specific datasets; it would be beneficial to see more diverse real-world applications to assess generalizability.
> >
>
> We are currently running experiments on the non-molecular IMDB dataset and will report the results next week.
>
> > W2: The answers to the questions asked are pretty obvious beforehand. The structural distances that stem from non-trainable graph kernels have nothing to do with the task, therefore it is unreasonable to assume that an MPNN (before or after training) would be highly correlated (Q2, Q3) . The same goes for Q4, Q5, where the functional distance encodes the target, and is therefore what the MPNN is optimized for. While it is not inherently bad to ask questions that one expects the answer to, these questions, though many, create little new insight.
> >
>
> While the analyses of $d_{MPNN}$ in Sections 3 and 4 may seem obvious, they offer different insights than previous studies on $d_{MPNN}$. In short, we found that the alignment between $d_{MPNN}$ and $d_{func}$ is more important to MPNN performance than the alignment between $d_{MPNN}$ and $d_{struc}$, which has been studied in previous works.
> However, we acknowledge your concerns and change the flow of the paper to spend less space on this analysis. We will move most of the content of Section 3 to appendix, and use the remaining space to explain in detail the theory and algorithm of WILT.
>
> > W3: The algorithm for learning the WILT weight is only discussed in the appendix.
> >
>
> We will include the explanation of the algorithm in the main text in the updated paper.
>
> > Q1: How expressive is WILT? It implies a hyperbolic distance between colors, so intuitively, it should be weaker than MPNNs?
> >
>
> In terms of binary expressive power, WILT is more expressive than MPNN:
>
> - MPNN (mean pooling) $\le$ WILT (size normalization) (Theorem 4)
> - MPNN (sum pooling) $\le$ WILT (dummy normalization) (Theorem 5)
>
> However, when it comes to how much different distance structure each distance can learn, it is expected that $d_{MPNN}$ can express more diverse distance structures than $d_{WILT}$. This is because $d_{WILT}$ is limited to an optimal transport on a tree of WL-colors, while an MPNN can place any WL-color in a d-dimensional Euclidean space. In practice, however, we have found that $d_{WILT}$ is expressive enough to model $d_{MPNN}$ (Section 6). Moreover, this restriction allows for the fast computation and interpretation of $d_{WILT}$ (Proposition 1).
>
> > Q2: How long does learning the WILT weights take?
> >
>
> In general, training is very efficient because the graph distance $d_{WILT}(G, H)$ can be computed  via a weighted Manhattan distance on suitable, precomputed vector embeddings of graphs, i.e., via tensor operations. We will include the practical running time in the appendix.
>
> > Q3: Famously WL is extremely sensitive to noise in the graph structure. Does WILT handle structural noise and/or feature noise well?
> >
>
> This is actually a very interesting question, that we did not consider, yet. Intuitively, WILT is more robust to structure/feature noise the than WL test, because WILT can adjust edge weights to account for such noise. The robustness to noise is beyond the scope of our current paper, but we will mention it as a future work.
>
> ---
>
> Thank you again for your insightful feedback. Please let us know if you have any further questions.

---

> > ### Comment · Reviewer_H8oL · 2024-11-25
> >
> > Dear Authors,
> >
> > Thank you for your detailed response. My questions have been answered and I maintain my score of 8.

---

### Author Response · Authors · 2024-11-15

Thank you for your detailed questions and suggestions. As some of you pointed out an unclear motivation and logical flow, we will adapt our paper to incorporate the reviewers suggestions and address the identified shortcomings. We now explain the motivation of our paper and what each section conveys. We will post an updated version of the paper pdf next week.

The high performance of MPNNs has mainly been explained and analyzed in terms of their binary expressive power. Recently, some studies have investigated non-binary expressiveness by analyzing the geometry of the MPNN embedding space [1, 2]. However, [1] just upper-bounded $d_{MPNN}$ with a task-irrelevant $d_{struc}$. [2] showed the equivalence of two *structural* pseudometrics on graphons and required the consideration of *all* MPNNs with some Lipschitz constant. This casts doubt on the applicability of these analyses to any particular MPNN trained on sparse graphs. Thus, our first research question is: What properties do $d_{MPNN}$ have in practice that can explain the high performance of MPNNs? We address this question in Sections 3 and 4, where we compare $d_{MPNN}$ with $d_{struc}$ and $d_{func}$, respectively. Here are the main findings in these sections:

- Although the previous studies have focused on the alignment between $d_{MPNN}$ and $d_{struc}$, it is not improved by training and is not strongly correlated with predictive performance.
- Rather, the alignment between $d_{MPNN}$ and $d_{func}$ improves strongly and consistently with training and is highly correlated with performance.

Thus, we need a different approach to understand $d_{MPNN}$, which leads to our second question: How do MPNNs learn such a metric structure that respects $d_{func}$? As MPNNs essentially view graphs as multisets of Weisfeiler Leman (WL) colors, we propose a method to identify which WL colors affect $d_{MPNN}$ most. Specifically, we distill MPNNs to WILT while preserving the graph distance (Section 5.4, Appendix C). The investigation of the resulting edge weights of WILT offers novel insights into the MPNN embedding space (Section 6, Appendix F):

- $d_{MPNN}$ is determined by only a small fraction(~5%) of the entire set of Weisfeiler Leman (WL) colors.
- The identified WL colors are also known to be important by domain experts.

In addition, our graph pseudometric $d_{WILT}$ has several desirable properties:

- $d_{WILT}$ is computable in linear time since it is an optimal transport on a tree (Proposition 1 in Section 5.2)
- $d_{WILT}$ generalizes well-known graph kernels (Section 5.3, Appendix B.3)
- $d_{WILT}$ has the same expressive power as the 1-WL test (Appendix B.4)

[1] Chuang, C. Y., & Jegelka, S. (2022). Tree mover's distance: Bridging graph metrics and stability of graph neural networks. *Advances in Neural Information Processing Systems*, *35*, 2944-2957.

[2] Böker, J., Levie, R., Huang, N., Villar, S., & Morris, C. (2024). Fine-grained expressivity of graph neural networks. *Advances in Neural Information Processing Systems*, *36*.

---

> ### Author Response · Authors · 2024-11-24
>
> Dear reviewers, we are sorry to have kept you waiting. We have uploaded an updated version of our paper. Here are the main updates.
>
> - Clarify the motivation and research question in the Abstract and Introduction section.
> - Move the analyses of the alignment between $d_{MPNN}$ and task-irrelevant $d_{struc}$ to Appendix E.
> - Add examples of how to compute the two types of normalized $d_{WILT}$ in Section 4.3.
> - Add the distillation algorithm to Algorithm 1 in Section 4.4.
> - Analyze the expressiveness of $d_{WILT}$ in Section 4.5.
> - Add experimental results on IMDB-BINARY and COLLAB datasets to Appendix D and E.
> - Add experimental results on IMDB-BINARY to Appendix F (experiments on COLLAB are in progress).
>
> If you still find something unclear, or have further questions, please do not hesitate to ask.

---

### Meta-Review · Area_Chair_1EcS · 2024-12-18

**Metareview:**

The authors consider a metric for message passing neural networks (MPNNs). The authors argue that the alignment between the distance for MPNNs and functional distance is more relevant than the alignment between the distance for MPNNs and structural distance. The authors propose Weisfeiler Leman Labeling Tree (WILT) for optimal transport, i.e., tree-Wasserstein, and exploit the closed-form expression of tree-Wasserstein for a fast computation.

The Reviewers have mixed opinions on the submission. The Reviewers agree that the proposed WILT distance is interesting with its fast computation. However, the Reviewers raised concerns about the binary nature of the considered distances which limits its expressivity, and leads to unconvincing comparison for using different metrics for evaluation. The Reviewers also raised concerns on the empirical evidences to support the claims, e.g., better alignment of distance of MPNNs to functional distance. Therefore, we think that the submission is not ready for publication yet. The authors may follow the Reviewers' comments to improve the submission.

**Additional Comments On Reviewer Discussion:**

The Reviewers have mixed opinions on the submission. The proposed WILT distance is interesting. However, the Reviewers raised concerns on the expressivity (e.g., binary distance), and empirical evidences (e.g., different metrics for evaluation).

---

### Decision · Program_Chairs · 2025-01-22

Reject